# On the differences in the vertical distribution of modeled aerosol optical depth over the southeast Atlantic

Ian Chang[1], Lan Gao[1], Connor J. Flynn[1], Yohei Shinozuka[2,3], Sarah J. Doherty[4,5], Michael S. Diamond[5,a,b], Karla M. Longo[6], Gonzalo A. Ferrada[7], Gregory R. Carmichael[7], Patricia Castellanos[8], Arlindo M. da Silva[8], Pablo E. Saide[9,10], Calvin Howes[10], Zhixin Xue[11,c], Marc Mallet[12], Ravi Govindaraju[13], Qiaoqiao Wang[14], Yafang Cheng[15], Yan Feng[16], Sharon P. Burton[17], Richard A. Ferrare[17], Samuel E. LeBlanc[2,3], Meloë S. Kacenelenbogen[8], Kristina Pistone[3,2], Michal Segal-Rozenhaimer[2,3,18], Kerry G. Meyer[8], Ju-Mee Ryoo[2,19], Leonhard Pfister[2], Adeyemi A. Adebiyi[20], Robert Wood[5], Paquita Zuidema[21], Sundar A. Christopher[11], Jens Redemann[1]

[1]School of Meteorology, University of Oklahoma, Norman, Oklahoma, USA
[2]NASA Ames Research Center, Moffett Field, California, USA
[3]Bay Area Environmental Research Institute, Moffett Field, California, USA
[4]Cooperative Institute for Climate, Ocean and Ecosystem Studies, University of Washington, Seattle, Washington, USA
[5]Department of Atmospheric Science, University of Washington, Seattle, WA, USA
[a]Now at NOAA Chemical Sciences Laboratory (CSL), Boulder, Colorado, USA
[b]Now at Cooperative Institute for Research in Environmental Sciences (CIRES), University of Colorado, Boulder, Colorado, USA
[6]National Institute for Space Research, São José dos Campos, Brazil
[7]Center for Global and Regional Environmental Research, University of Iowa, Iowa City, Iowa, USA
[8]NASA Goddard Space Flight Center, Greenbelt, Maryland, USA
[9]Institute of the Environment and Sustainability, University of California, Los Angeles, Los Angeles, California, USA
[10]Department of Atmospheric and Oceanic Sciences, University of California, Los Angeles, Los Angeles, California, USA
[11]Department of Atmospheric and Earth Science, University of Alabama in Huntsville, Huntsville, Alabama, USA
[c]Center for Global and Regional Environmental Research, University of Iowa, Iowa City, Iowa, USA
[12]Centre National de Recherches Météorologiques, UMR3589, Météo-France-CNRS, Toulouse, France
[13]Science Systems and Applications, Inc., Greenbelt, Maryland, USA
[14]Institute for Environmental and Climate Research, Jinan University, 510632 Guangzhou, China
[15]Minerva Research Group, Max Planck Institute for Chemistry, 55128 Mainz, Germany
[16]Environmental Science Division, Argonne National Laboratory, Argonne, Illinois, USA
[17]NASA Langley Research Center, Hampton, Virginia, USA
[18]Department of Geophysics, Porter School of the Environment and Earth Sciences, Tel-Aviv University, Israel
[19]Science and Technology Corporation (STC), Moffett Field, CA, USA
[20]Department of Life and Environmental Sciences, University of California, Merced, Merced, California, USA
[21]Rosenstiel School of Marine, Atmospheric, and Earth Science, University of Miami, Miami, Florida, USA

*Correspondence to*: Ian Chang (ian.chang@ou.edu) and Lan Gao (lgao@ou.edu)

**Abstract.** The southeast Atlantic is home to an expansive smoke aerosol plume overlying a large cloud deck for approximately a third of the year. The aerosol plume is mainly attributed to the extensive biomass burning activity that occurs in southern Africa. Current Earth system models (ESMs) reveal significant differences in their estimates of regional aerosol radiative effects over this region. Such large differences partially stem from uncertainties in the vertical distribution of aerosols in the

troposphere. These uncertainties translate into different aerosol optical depths (AOD) in the planetary boundary layer (PBL) and the free troposphere (FT). This study examines differences of AOD fraction in the FT and AOD differences among ESMs (WRF-CAM5, WRF-FINN, GEOS-Chem, EAM-E3SM, ALADIN, GEOS-FP, and MERRA-2) and aircraft-based

measurements from the NASA ObseRvations of Aerosols above CLouds and their intEractionS (ORACLES) field campaign. Models frequently define the PBL as the well-mixed surface-based layer, but this definition misses the upper parts of decoupled PBLs, in which most low-level clouds occur. To account for the presence of decoupled boundary layers in the models, the height of maximum vertical gradient of specific humidity profiles from each model is used to define PBL heights.

Results indicate that the monthly mean contribution of AOD in the FT to the total-column AOD ranges from 44% to 74% in September 2016 and from 54% to 71% in August 2017 within the region bounded by 25°S – 0° and 15°W – 15°E (excluding land) among the ESMs. ALADIN and GEOS-Chem show similar aerosol plume patterns to a derived above-cloud aerosol product from the Moderate Resolution Imaging Spectroradiometer (MODIS) during September 2016, but none of the models show a similar above-cloud plume pattern as MODIS does in August 2017. Using the second-generation High Spectral

Resolution Lidar (HSRL-2) to derive an aircraft-based constraint on the AOD and the fractional AOD, we found that WRF-CAM5 produces 40% less AOD than those from the HSRL-2 measurements, but it performs well at separating AOD fraction between the FT and the PBL. AOD fractions in the FT for GEOS-Chem and EAM-E3SM are, respectively, 10% and 15% lower than the AOD fractions from the HSRL-2. Their similar mean AODs reflect a cancellation of high and low AOD biases. Compared with aircraft-based observations, GEOS-FP, MERRA-2, and ALADIN produce 24% - 36% less AOD and tend to

misplace more aerosols in the PBL. The models generally underestimate AODs for measured AODs that are above 0.8, indicating their limitations at reproducing high AODs. The differences in the absolute AOD, FT AOD, and the vertical apportioning of AOD in different models highlight the need to continue improving the accuracy of modeled AOD distributions. These differences affect the sign and magnitude of the net aerosol radiative forcing, especially when aerosols are in contact with clouds.


## 1 Introduction

Estimates of aerosol radiative effects in Earth system models (ESMs) reveal large differences (e.g., Stier et al., 2013; Myhre et al., 2013, 2017; Bellouin et al., 2020; Myhre et al., 2020), particularly at the regional scale (Haywood et al., 2020). This is important because aerosol-radiation interactions and aerosol-cloud interactions contribute significant uncertainties to total

anthropogenic forcing (Forster et al., 2021). Uncertainties in regional aerosol radiative effects over the southeast Atlantic, for example, are attributed to biases in modeled aerosol spatial distributions, aerosol absorption, and cloud fraction stemming from differences in modeling approaches and parameterizations (Mallet et al., 2021; Doherty et al., 2022). When aerosols are present within clouds, aerosol-cloud microphysical interactions can produce forcing by altering cloud reflectivity and lifetime (Twomey, 1974; Albrecht, 1989; Costantino and Bréon, 2013). In the absence of physical interactions with clouds, aerosols

can alter the global and regional radiation budget via the direct aerosol radiative effects (Feng and Christopher, 2015; Chang and Christopher, 2017; Kacenelenbogen et al., 2019; Thorsen et al., 2020) and semi-direct effects (Johnson et al., 2004; Koch and Del Genio, 2010; Sakaeda et al., 2011; Zhang and Zuidema, 2019; Deaconu et al., 2019; Das et al., 2020; Herbert et al., 2020; Zhang and Zuidema, 2021). Thus, accurate modeling of aerosol composition, optical properties, and spatial distributions, both vertically and horizontally, is crucial for accurate estimates of aerosol radiative effects.


During austral spring, high loadings of biomass burning smoke aerosols are present above semi-permanent stratocumulus clouds over the southeast Atlantic (Adebiyi et al., 2015; Chang and Christopher, 2016; Zuidema et al., 2016; Haywood et al., 2021; Redemann et al., 2021). The true color satellite image captured by the Moderate Resolution Imaging Spectroradiometer (MODIS) instrument in Figure 1a shows aerosols over the southeast Atlantic Ocean and widespread fire activities over sub-
Saharan Africa, the latter indicated by orange dots symbolizing individual fire sources. Stratocumulus clouds appear slightly darkened over the ocean due to the attenuation of cloud reflection by the smoke aerosol. Figures 1b and 1c show the monthly mean above-cloud aerosol optical depth (ACAOD) as derived using a retrieval algorithm that accounts for above-cloud absorbing aerosol (Meyer et al., 2015), applied to MODIS (combined Terra and Aqua) for September 2016 and August 2017, respectively. The fire  frequency plots are derived from the MODIS Collection 6 fire product (MXD14) (Giglio et al., 2016)
over southern Africa.

The vertical distribution of aerosol plays an important role in determining the outcome of aerosol-cloud-radiation interactions (Koch and Del Genio, 2010; Das et al., 2017). Even without the presence of clouds, accurate modeling of the aerosol optical depth (AOD) is crucial since AOD biases are responsible for about 25% of the clear-sky top-of-atmosphere (TOA) shortwave
flux biases between 60°S and 60°N over the global oceans (Su et al., 2021). Given this, the Aerosol Comparisons between Observations and Models (AeroCom) project has provided comprehensive aerosol evaluations of ESMs against observations (Koffi et al., 2012; Textor et al., 2006). Shinozuka et al. (2020) compared the apportionment of aerosol optical properties in the free troposphere (FT) and planetary boundary layer (PBL) from various models over the southeast Atlantic, and they found that the ratio of FT to PBL AOD are affected by the differences across models in their definition of PBL height. However,
their studies were limited to the ORACLES 2016 field campaign and along the designated routine flight tracks. Given that aerosol properties in models vary significantly both horizontally and vertically (e.g., Doherty et al., 2022), the partitioning of layer-integrated quantities such as AODs in the FT and in the PBL will also differ significantly across ESMs. In contrast, the present study examines AOD partitioning from both ORACLES 2016 and ORACLES 2017 field campaign since the differences in the multi-year apportionment of AOD in the FT and PBL in various models over the southeast Atlantic remain
largely unexplored.

The main objective of this study is to identify the proportion of AOD within the FT relative to the total-column (i.e., FT plus PBL) AOD in ESMs and in aircraft-based lidar measurements during the NASA ObseRvations of Aerosols above CLouds and

their intEractionS (ORACLES) field experiment. Such an analysis provides a perspective on how much aerosol loadings are

potentially interacting with different cloud phases such as low-level (below 3 km) clouds and mid-level (between 3 km and 8 km) clouds. While low-level clouds are the predominant cloud type in the southeast Atlantic during the biomass burning season, mid-level clouds can also be present and be in contact with aerosols above low-level liquid clouds (Adebiyi et al., 2020). Furthermore, the apportioning of AOD to the FT and the PBL can influence the relative roles of aerosol direct, semi-direct, and indirect forcing, which affects the sign and magnitude of aerosol climate forcing. Observational-based studies using

the spaceborne Cloud-Aerosol Lidar with Orthogonal Polarization (CALIOP) had shown that the FT has relatively higher AOD than the PBL AOD over the southeast Atlantic (Bourgeois et al., 2018; Painemal et al., 2019). However, CALIOP often misses more tenuous aerosols than aircraft-based lidars (e.g., Kacenelenbogen et al., 2011; Winker et al., 2013). Another objective of this study is to evaluate AODs from models against those from aircraft measurements, including measurements from the second-generation High Spectral Resolution Lidar (HSRL-2) (Burton et al., 2018; Hair et al., 2008) and the NASA

Ames Spectrometers for Sky-Scanning, Sun-Tracking Atmospheric Research (4STAR) (Dunagan et al., 2013).

## 2 Data and methods

### 2.1 The NASA ORACLES field campaign

The NASA ORACLES project was conducted to pursue an unprecedented investigation of aerosol-cloud-radiation interactions between smoke aerosols and stratocumulus clouds during the late austral spring in the southeast Atlantic (Redemann et al.,

2021). Several other international field experiments were conducted in this region during the same period, providing synergistic field measurements (Formenti et al., 2019; Haywood et al., 2021; Zuidema et al., 2016, 2018). The ORACLES field campaign utilized the NASA P-3 aircraft to make measurements based out of Walvis Bay, Namibia in September 2016 and São Tomé and Príncipe in August 2017 and September/October 2018 (for a total of about 350 science flight hours). In 2016, the NASA ER-2 aircraft augmented the field campaign with remote sensing measurements, adding approximately 100 science flight

hours. ORACLES adopted a systematic sampling strategy for one-half of its flights, in which the same track was repeatedly sampled without consideration of meteorology. These flights are representative of the monthly-mean in their totality (e.g., Shinozuka et al., 2020b; Doherty et al., 2022). The present study focuses on the ORACLES 2016 and 2017 field campaigns, when a similar number of ESM simulations are available. Figure 2 shows the locations of the AOD measurements acquired during the ORACLES 2016 and 2017 that are used to evaluate modeled AODs in this study.

**2.2 Descriptions of models and data assimilation systems**

We evaluate seven ESMs using both clear-sky and above-cloud AOD measured during ORACLES 2016 and five ESMs using data from ORACLES 2017. The treatment of aerosol processes and the assumed microphysical and optical properties per species are significantly different among the ESMs. Tables

Table 1 describes the grid resolution, process schemes, meteorological parameters, emission sources, and other key features of each model. Modern-Era Retrospective-analysis for Research and Applications- Version 2 (MERRA-2) was developed at NASA's Global Modeling and Assimilation Office (GMAO) (Gelaro et al., 2017; Randles et al., 2017) using the three-dimensional variational data assimilation Gridpoint Statistical Interpolation (GSI) meteorological analysis scheme (Wu et al., 2002; Kleist et al., 2009). The Goddard Chemistry Aerosol Radiation and Transport (GOCART) aerosol module assumes five externally mixed aerosol species, and it is coupled to a radiation parameterization. Sulfate, organic carbon (OC), and black carbon (BC) are represented by lognormal distributions with fixed dry aerosol mean diameter and standard deviation, while dust and sea salt distributions are resolved by five size bins. The aerosol assimilation is based on satellite clear-sky AODs derived from a neural network retrieval (NNR) approach (Buchard et al., 2015; Randles et al., 2017).

We also examine Version 4.2.2 of Weather Research and Forecasting model coupled with chemistry (WRF-Chem) using biomass burning emissions from Version 2.4 of the Fire INventory from NCAR (FINN) emission (hereinafter WRF-FINN). FINNv2.4 merges fire detection data from both MODIS and the Visible Infrared Imaging Radiometer Suite (VIIRS) satellite sensors, increasing the areal coverage of the actual burned areas relative to the previous versions. Meteorological initial and lateral boundary conditions for WRF-FINN simulations are obtained from ERA5 (Hersbach et al., 2020). The Morrison two-moment cloud microphysical scheme and the Model for Simulating Aerosol Interactions and Chemistry (MOSAIC) mechanism are adopted to simulate the aerosol-cloud interactions (Morrison et al., 2005; Zaveri et al., 2008; Zaveri and Peters, 1999). The MOSAIC four-bin aerosol module is coupled with the Model for Ozone and Related chemical Tracers (MOZART) (Emmons et al., 2010) gas phase chemical scheme (Knote et al., 2014). This model uses the ambient relative humidity to account for hygroscopic growth. Here, the preliminary version of the MOZART-T1 (MOZART tropospheric) scheme was used that does not include a detailed treatment of monoterpenes, MBO, aromatics, HONO, $C_2H_2$, and uses a new oxidation scheme (Hodzic and Knote, 2014). The description of the complete MOZART-T1 version is documented in Emmons et al. (2020).

The remaining five models in this study were evaluated by Shinozuka et al. (2020); these include WRF-Chem that couples with the Community Atmosphere Model-Version 5 (WRF-CAM5), the French Aire Limitée Adaptation dynamique Développement Inter-National (ALADIN) climate model, the Goddard Earth Observing System-Forward Processing (GEOS-FP, previously known as GEOS-5), GEOS-Chem, and Version 1 of the Energy Exascale Earth System Model (E3SM) Atmosphere Model (hereinafter EAM-E3SM) of United States Department of Energy (DOE). Aerosol optical properties in the WRF-Chem configurations are computed using Mie theory code and Chebyshev expansion coefficients for pre-specified aerosol size bins. The tri-modal version of the Modal Aerosol Module (MAM3) (Liu et al., 2012) in CAM5 is used assuming internal mixture within lognormal modes and a volume mixing rule (Fast et al., 2006). The ALADIN smoke aerosol optical properties are assumed to be externally mixed with an imaginary refractive index of 0.03 (at 550 nm) for both fresh and aged smoke following a fixed lognormal size distribution (Mallet et al., 2019, 2020). Aerosol optical properties in

GEOS-Chem assume externally mixed aerosol (Koepke et al., 1997), with aerosol particle sizes assumed to follow a lognormal size distribution (Wang, 2003). For EAM-E3SM, the aerosol optical properties are assumed to be internally mixed within three size modes (Aitken, accumulation, and coarse) and aerosol hygroscopic growth is accounted for as described by Ghan and Zaveri (2007). This model includes an extra primary carbon mode to represent freshly-emitted primary organic matter and black carbon (Liu et al., 2016; Wang et al., 2020). In this study, the 2016 EAM-E3SM model is based on the 2016 meteorology from the ECMWF reanalysis rather than the free-running meteorology as in Shinozuka et al. (2020), which would imply a better simulation of aerosol transport.

GEOS-FP and MERRA-2 are the only models in this study that uses AOD assimilation. MERRA-2 is based on Version 5.12.4 of GEOS. GEOS-FP assimilates observed AODs from satellite and ground-based measurements whereas MERRA-2 only assimilates satellite AODs and does not assimilate ground-based AODs during the study period. While both assimilation systems use the relaxed Arakawa–Schubert convective parameterization (Moorthi and Suarez, 1992), MERRA-2 includes a precipitation correction algorithm that modulates the aerosol wet deposition differently than GEOS-FP (Reichle et al., 2017). Another difference between these two systems is that MERRA-2 was ran at 0.5° resolution whereas GEOS-FP was ran at a 0.25° resolution for September 2016 and at 0.125° resolution in August 2017.

### 2.3 The Second-generation High Spectral Resolution Lidar (HSRL-2)

The HSRL-2 directly measures vertical profiles of molecular and aerosol backscattering coefficients (at 355 nm and 532 nm), obviating the need for an inversion algorithm that assumes a lidar ratio (i.e., the ratio of aerosol backscattering to extinction) (Burton et al., 2018; Hair et al., 2008). The main difference between HSRL-2 and its predecessor HSRL-1 is the additional 355 nm channel. This downward-pointing lidar also measures the attenuated aerosol backscatter at 1064 nm and particle depolarization ratios at 355 nm, 532 nm, and 1064 nm. The HSRL-2 extinction profile is derived from the measured attenuated molecular backscattering profile, by isolating the attenuation due to aerosol extinction by comparison with the un-attenuated molecular backscatter profile derived with very small uncertainty from MERRA-2's molecular density profiles. During ORACLES 2016, the HSRL-2 was deployed on the NASA ER-2 aircraft, which typically flew at 20 km altitude. Therefore, it observed profiles of aerosols and clouds through the entire troposphere. In 2017, the low-flying P-3 aircraft carried the HSRL-2. Moreover, for the first 1,500 m below the aircraft, HSRL-2 does not report backscatter due to incomplete overlap between the laser and the telescope. We use the layer-accumulated AOD product from the highest altitude with valid backscatter measurements below the aircraft down to 50 meters above cloud-top height for above-cloud AOD conditions and the full-column for cloud-free conditions. This 50-meter buffer is implemented to minimize ambiguity associated with the transition at the cloud top from hydrated aerosol to cloud (Shinozuka et al., 2020a). Hence, the vertical extent of comparison between HSRL-2 and models is substantially shallower in 2017 than in 2016.

**2.4 Spectrometers for Sky-Scanning, Sun-Tracking Atmospheric Research (4STAR)**

The 4STAR instrument (Dunagan et al., 2013) flew aboard the NASA P-3 aircraft during ORACLES. 4STAR is an airborne sunphotometer that measures the hyperspectral direct solar beam transmittance between 350 and 1700 nm with a spectral resolution of 2 – 3 nm for the 350 – 1000 nm spectral range and 3 – 7 nm for the 1000 – 1700 nm spectral range. The measurements are converted to above-aircraft columnar AOD (Shinozuka et al., 2013; LeBlanc et al., 2020). The instrument also has capabilities to retrieve trace gas column concentration (Segal-Rosenheimer et al., 2014), aerosol intensive properties

such as single scattering albedo (SSA) from sky radiance measurements (Pistone et al., 2019), and cloud properties from cloud transmittances (LeBlanc et al., 2015). LeBlanc et al. (2020) discussed the necessary calibrations and corrections to obtain the 4STAR AOD during ORACLES. This data set contains either the above-cloud AOD or the full-column AOD, as indicated by a flag. This study compares the highest quality-assured 4STAR ACAOD data (at 550 nm) to collocated layer-integrated AOD from the ESMs over the same range of altitude. This ACAOD flag is created by manually inspecting aircraft vertical profiles

for changes in AOD and in situ scattering coefficient measurements above clouds. These clouds were identified during vertical profiling near the ACAOD measurements. They were defined by a cloud drop concentration exceeding 10 cm$^{-3}$ as measured by the Artium Flight Probe Dual Range Phase Doppler Interferometer (PDI). Details of the 4STAR ACAOD flag are described in LeBlanc et al. (2020).

**2.5 MODIS above-cloud aerosol satellite observations**

We use the ACAOD (i.e., MXD06ACAERO) product (Meyer et al., 2015) from the MODIS instruments on board the Aqua and Terra satellites to qualitatively compare the aerosol plume patterns between the observed and modeled AOD in the FT. This above-cloud AOD product is used instead of the standard MODIS AOD (i.e., MXD04) product in the comparisons with modeled AOD because the latter only performs AOD retrievals in clear-sky (i.e., cloud-free) areas. The above-cloud AOD

product utilizes reflectances from six solar spectral channels (0.47, 0.55, 0.66, 0.87, 1.24, and 2.13µm) to simultaneously retrieve the above-cloud AOD and the underlying cloud optical depth. The retrieval algorithm assumes the absorbing model of the MODIS Dark Target land aerosol product (Levy et al., 2009). This product tends to retrieve higher ACAOD compared to HSRL-2 and 4STAR during ORACLES 2016, with a mean bias error of 0.07 and 0.12, respectively (Chang et al., 2021). The assumed SSA (i.e., 0.87 at 550 nm) in the MODIS retrieval is above the 90[th] percentile of the observed SSA retrieved

from 4STAR observations during the ORACLES 2016 deployment. The 4STAR retrieved a median SSA of about 0.84 during September 2016, so the higher assumed SSA contributed to the higher MODIS ACAOD retrieval. Comprehensive statistical evaluations of the ACAOD retrievals against aircraft measurements are presented in Chang et al. (2021).

**2.6 Computation of planetary boundary layer heights**

The PBL is the layer of the atmosphere where atmospheric properties directly interact with and are influenced by the surface (Seidel et al., 2010). Over oceans, the PBL deepens with increasing sea surface temperatures, promoting its decoupling and deepening (Wood and Bretherton, 2004). Stratocumulus clouds often occur in the upper part of decoupled PBLs in the

southeast Atlantic, and the PBL height tends to increase away from the southwest African coast before transitioning to a cumulus-dominated cloud regime (Zhang and Zuidema, 2019; Ryoo et al., 2021; Zhang and Zuidema, 2021). Models

frequently define the PBL as the surface well-mixed layer, but this definition misses the upper parts of decoupled PBLs, in which most low-level clouds occur. This exclusion leads to an underestimated PBL height and poor correlation between the top of the model-defined PBL and the low cloud-top height. Thus, we apply an alternative method of estimating PBL height that includes decoupled stratocumulus clouds that are above the surface mixed layer using profiles of specific humidity, $q$ (Ryoo et al., 2022). The $q$-inferred PBL height tends to be from several hundred meters to a few kilometers higher than the top

of the surface mixed layer. In our analysis, layers above this definition of PBL are considered to be in the FT. Comparisons of $q$-inferred PBL height from the models and HSRL-2's cloud-top height (CTH) during ORACLES 2016 are presented in Figures S 2 to S 8 using the mean absolute error (*MAE*) and the mean bias error (*MBE*) (Simon et al., 2012):

$$MAE = \frac{1}{N}\sum|observed\ CTH - modeled\ PBLH|\ , (1)$$

$$MBE = \frac{1}{N}\sum(observed\ CTH - modeled\ PBLH)\ , (2)$$

Modeled PBL heights derived this way tend to be higher than collocated CTHs from HSRL-2, with MBE ranging between -6 m and -514 m. EAM-E3SM's maximum PBL height only reaches 1,560 m and ALADIN's minimum PBL is 720 m. The 2017 comparisons (Figures S 9 to S 14) are for locations further north and west than the locations of comparison in 2016, so the PBL height is generally higher. Overall, the 2017 comparisons have larger differences than the 2016 comparisons, with the MBE ranging between -414 m and -1,037 m, indicating that the models tend to position the PBL height higher than they should

away from the coast. Note that the main objective of the present study focuses on how each model partitions AOD in the FT compared to the PBL, in the context of the PBL definition above. The PBL biases based on each model's original definition of PBL height and their impact on partitioning AOD in these two layers among the models requires a separate investigation.

**2.7 Aircraft-model AOD intercomparison methods**

Before evaluating modeled AOD, we first spatially and temporally interpolate modeled AODs using linear interpolation to the

exact location and time of each aircraft measurement. For the HSRL-2 measurements, we distinguish their AOD measurements as either a FT or a clear-sky column depending on whether clouds are present in the column of interest. Hence, HSRL-2 columns are assigned to a FT evaluation when clouds are present and to a clear-sky evaluation in the absence of clouds. HSRL-2 AOD is used in two parts of analysis: 1) the statistical distribution analysis of AOD fractions in the FT and AODs between the models and the HSRL-2 and 2) the instantaneous evaluation of modeled AODs against aircraft measurements. In the first

analysis, the modeled PBLs are used to partition AODs in the FT from the PBL. We only use the clear-sky data in the HSRL-2 measurements since partitioning AOD between the FT and the PBL is possible only when the HSRL-2 does not identify cloud presence (i.e., a cloud-free condition) below the instrument.

In all comparisons throughout this study, we compare modeled AODs with aircraft-based AODs over the same altitudinal

ranges. In FT AOD comparisons, whether a full-column or partial-column modeled FT AOD is evaluated against the HSRL-2 depends on the modeled PBL height relative to HSRL-2 CTH and whether the HSRL-2 is carried on the ER-2 or the P-3. In the 2016 comparison, the entire modeled AOD in the FT is evaluated against HSRL-2's above-cloud AOD if the modeled PBL height is higher than HSRL-2's CTH in that column. In those cases, we only consider HSRL-2's aerosol layer from the ER-2 altitude down to the modeled PBL height in order to compare AOD for the same physical thickness. In contrast, if the modeled

PBL height is lower, we only consider the modeled aerosol layer down to the altitude that the HSRL-2 indicates as the CTH. Since the HSRL-2 flew on the ER-2 at about 20 km altitude during September 2016 and on the lower-flying P-3 aircraft (maximum altitude of about 6 km) in August 2017, we generally compare HSRL-2 AOD over a larger vertical column in September 2016 than in August 2017 within each 1° horizontal grid. Moreover, the first 1,500 m gap below the aircraft means that AODs are measured over a shorter vertical distance than the distance from the aircraft to the cloud top. For this reason,

we only consider HSRL-2 data when the P-3 flew above 5,000 m so that we could attain data from at least 3,500 meters down to the cloud top in August 2017.

Aerosols in model layers where clouds are also present include extinction from hydrated aerosol, which would cause a higher AOD than it would otherwise without clouds (Quaas et al., 2009; Neubauer et al., 2017). Comparisons between AOD

measurements from HSRL-2 and modeled AOD exclude modeled layers where clouds are present and exclude AOD measurements for those layers in the HSRL-2 as well. Above-cloud AOD measurements from 4STAR and HSRL-2 showed a strong agreement when they were collocated within 15 minutes at the same location (Chang et al., 2021), so systematic AOD biases in either instrument are unlikely. However, 4STAR measurements only provide the above aircraft column AOD, equivalent to total column AOD when sampling from low altitude. Thus, 4STAR is unsuitable for a layer-selective comparison

because transmission-based aerosol measurements can only offer the altitude-resolved AOD during vertical profiling and hence cannot provide AOD over a sublayer. While layered AODs can be derived, they require a combination of measurements in time and space (Shinozuka et al., 2011; LeBlanc et al., 2020), limiting data availability, so they are not used in this study. Given these limitations, AOD comparisons between 4STAR and models include all the modeled layers above the P-3 altitude regardless of cloud presence at specific model layers.


In the second analysis, we evaluate the models' performances using various statistical metrics. We aggregate modeled and aircraft AODs to 1° grid resolution, which is approximately the median native grid resolution of the ESMs that we examine in this study. The Spearman's Rank correlation coefficient is used instead of the Pearson's linear correlation coefficient since the former is statistically less sensitive to outliers (Sayer et al., 2019; Sayer, 2020). We also evaluate the *RMSE*, the fractional

error (*FE*), and the fractional bias (*FB*):

$$FE = \frac{2}{N} \sum \frac{|modeled\ AOD - observed\ AOD|}{(modeled\ AOD + observed\ AOD)}, (3)$$

$$FB = \frac{2}{N} \sum \frac{(modeled\ AOD - observed\ AOD)}{(modeled\ AOD + observed\ AOD)} , (4)$$

where $N$ is the sample size. Note that $FB$ is similar to the relative mean bias reported by Shinozuka et al. (2020b) except for the addition of the modeled values in the denominator and the factor of two outside the summation. Typically, up to 100 points of aircraft data are averaged into a 1° grid box. Varying the aggregated grid resolution mainly affects standard deviations and has a very minor influence on other statistics such as correlations and root mean square error ($RMSE$). The $FE$ and $FB$ for each model agrees to within 0.04 of their respective value when the data are aggregated between grid resolutions of 0.25° to 2.5°, except for ALADIN where its $FE$ decreases by 0.09 in going from 0.25° to 2.5° grid resolution.

## 3 Results

### 3.1 Contributions of FT AOD from models and aircraft observations

The vertical distribution of aerosols affects the relative roles of the aerosol direct, semi-direct, and indirect forcing. It also relates to the amount of aerosol loading that can be lost to scavenging and entrainment, so it is useful to assess the relative amount of aerosol loadings that are in the FT and in the PBL. To examine the contributions of AOD (at 550 nm) in the FT to the total-column AOD, we compute the ratio of AOD in the FT to the total-column AOD for each model. Figure 3 shows the fraction of the FT AOD to the full-column AOD for September 2016. The AODs in GEOS-Chem and EAM-E3SM predominantly reside in the FT. The FT fraction of AOD in the other models generally decreases northwestward, which is consistent with PBL deepening and overall plume subsidence during transport in that direction. WRF-FINN's high fraction of FT AOD covers most of the southeast Atlantic south of 15°S whereas WRF-CAM5 has a high fraction of FT AOD only near coastal Namibia. WRF-CAM5, MERRA-2, and GEOS-FP have peaks in the FT AOD fraction off coastal Namibia, decreasing northwestward from the coast. MERRA-2 has over half the AOD in the FT for the majority of the southeast Atlantic, whereas GEOS-FP only has a high fraction of AOD in the FT south of 10°S. In ALADIN, the fractional AOD in the FT peaks at 13°S 6°E, with a shallower gradient decrease in the northwest-southeast direction than in southwest-northeast direction. A comparison of the modeled AOD fraction in the FT bounded by 25°S – 0° and 15°W – 15°E (excluding land) is summarized by a box-whisker plot in the bottom panel of Figure 3. The mean ratio ranges from 44% to 74% in September 2016. WRF-CAM5 has the lowest average fraction of AOD in the FT. Both WRF-FINN and ALADIN have a large spread in the fraction of FT AOD since their ratios are high in the dense stratocumulus region but drop sharply outside of it.

For August 2017, the high fraction of FT AOD extends further northwest (Figure 4) than in September 2016. WRF-FINN has a steeper northwestward gradient in the FT fraction of AOD than WRF-CAM5. GEOS-FP has a lower fraction of AOD in the FT than MERRA-2 whereas GEOS-Chem has a high fraction of AOD in the FT for most parts of the region. The box-whisker plot indicates a mean ratio ranging between 54% and 71%, which is narrower than the mean ratio range in September 2016 and corroborates with the northwest extension of high FT AOD fractions. WRF-FINN has the largest range of AOD fraction in the FT among the five models, which is consistent with its steepest ratio decline relative to other models.

An observation-based fractional AOD in the FT can be inferred from HSRL-2 clear-sky AOD measurements using modeled PBL height to separate the FT and the PBL. Figure 5 is a box-whisker plot showing HSRL-2's AOD fraction in the FT based on each modeled PBL height (in cyan) and the modeled AOD fraction in the FT at the same locations (in pink) during September 2016. The full-column AOD and the FT AOD for the HSRL-2 (in blue) and the models (in red) are also shown. Since the definition of PBL height is model-dependent, the HSRL-derived AOD fraction in the FT is different for the

comparison to each model. WRF-FINN's mean AOD fractions in the FT and AOD has the closest agreement among all the models. The mean AOD fraction in the FT for the HSRL-2 and WRF-CAM5 is similar to each other, but both the mean AOD and the FT AOD are about 40% lower than the mean AOD and AOD in HSRL-2. Thus, while WRF-CAM5 separates AOD in the FT and in the PBL reasonably well, it underreports AOD compared to aircraft-based measurements. In contrast, GEOS-Chem and EAM-E3SM have similar AOD and FT AOD to the HSRL-2 measurements, but their AOD fractions in the FT are

lower than HSRL-2's AOD fraction by about 10% and 15%, respectively. The AOD fractions in the FT for GEOS-FP, MERRA-2, and ALADIN are 10%-15% lower than those computed from the HSRL-2. Moreover, the AODs in these three models are 24%-36% lower than those measured from the HSRL-2. The modeled FT AODs are approximately 35% lower than FT AOD in the HSRL-2. This finding suggest that not only do these three models produce less aerosol loading, they tend to displace more aerosols in the PBL than the HSRL-2. For September 2016, LeBlanc et al. (2020) reported a ratio of the

above-cloud AOD to the total-column AOD (at 501 nm) of 0.89 from 4STAR, which is representative of the more limited spatial range of the P-3. Specifically, their statistics were mostly within the plume with high aerosol loading, similar to regions of high AOD ratios from the models in Figure 3. We exclude the August 2017 comparison since the HSRL-2 could not capture the entire column from the P-3 for a suitable analysis. The differences in the AOD ratios and AODs between aircraft-based observations and models reveal the significant differences in how ESMs represent contributions of FT AOD to the full-column

in addition to AOD differences. More detailed evaluations of the modeled AOD against aircraft-based observations are presented in Section 3.4.

## 3.2 Full-column AOD

The monthly mean full-column AODs for the seven models during September 2016 are shown in Figure 6. Near coastal southern Africa, WRF-CAM5 has lower AODs and weaker longitudinal variations of AODs compared to the other models. In

contrast, ALADIN, GEOS-FP, GEOS-Chem, EAM-E3SM, and MERRA-2 show strong AOD peaks near the coast, with AOD dropping rapidly westward. WRF-FINN has smaller longitudinal variations of AOD than other models except for WRF-CAM5. Differences in the biomass burning emissions used in the models (Figure S1) can provide some insight to possible causes of the different AODs in the models. QFED generates the most OC+BC among the three emission inventories used in this study. GFED has the lowest OC+BC emission with less than half of those in QFED whereas FINNv2.4's OC+BC

emissions are in between those two inventories but are closer to the QFED emission. WRF-FINN has the highest aerosol loading among the models. ALADIN and EAM-E3SM are based on the GFED emission, but they do not have a significantly

lower AOD than models that use the QFED emission such as WRF-CAM5, GEOS-FP, MERRA-2, and GEOS-Chem. Thus, the magnitude of carbonaceous aerosol emissions are clearly not the only factor dictating the downwind AOD.

The monthly mean full-column AOD for the five models in August 2017 is shown in Figure 7. AODs over the southeast Atlantic are larger in August 2017 than in September 2016, consistent with the satellite derived above-cloud AOD in Figure 1. Aerosol plumes are shifted northward in all models relative to September 2016 because emissions are typically further north during the early part of the burning season (Haywood et al., 2008; Redemann et al., 2021). Moreover, the peak southern Africa Easterly Jets (AEJs) occur further north in August than in September (Ryoo et al., 2021). Similar to September 2016, both

WRF-CAM5 and WRF-FINN have the elevated AOD throughout the northern domain, especially in WRF-FINN. MERRA's AOD plume extends further west than GEOS-FP's whereas GEOS-Chem has significantly more elevated AODs near the coast.

### 3.3 Free tropospheric AOD

The monthly mean free tropospheric AOD for September 2016 from the ESMs is shown in Figure 8. Near coastal Angola, GEOS-Chem has the highest AOD among the models. With a peak total-column AOD of only 0.5 near the coast, WRF-

CAM5's FT AOD only reaches 0.3. WRF-FINN and EAM-E3SM have the furthest north extent of aerosol plumes in the FT. GEOS-FP's FT AODs are lower than those in MERRA-2, which is consistent with the full-column comparisons. ALADIN and GEOS-Chem show similarity in aerosol plume patterns as the MODIS above-cloud aerosol plume. Note that this MODIS ACAOD product, however, tends to be higher than 4STAR and HSRL-2 AOD measurements, with a mean bias error of 0.12 and 0.07, respectively (Chang et al., 2021).


Modeled PBL heights generally increase northward and westward from the coast, with the exception of EAM-E3SM where the PBL height increases southward. WRF-CAM5, GEOS-FP, GEOS-Chem, and MERRA-2 have similar PBL patterns. PBL heights in WRF-FINN and ALADIN are lower than these four models. It is clear from the results that both PBL heights and vertical distributions of aerosols affect the FT AOD. Similar to the full-column AOD, the northward shift of the AOD from

September 2016 to August 2017 is also evident in the FT (Figure 9). The FT AOD comparison shows that the plume extends furthest west in WRF-FINN than in other models whereas GEOS-Chem has the largest aerosol loading near the coast. None of the models has spatial distributions of the FT AOD that closely resemble those of the MODIS above-cloud AOD.

### 3.4 Evaluation of the modeled full-column AOD against aircraft AOD

Figure 10 shows the total-column AOD comparisons between the models and the aircraft-based HSRL-2 observations in

September 2016. WRF-CAM5, GEOS-FP, MERRA-2, and ALADIN are biased low, with *FB*s ranging from -0.60 to -0.38. These models also show lower mean AODs compared with the HSRL-2 as indicated in Figure 5. GEOS-Chem produces lower AODs for low HSRL-2 AODs and higher AODs for HSRL-2 AODs, resulting in an *FB* and *FE* of 0 and 0.36, respectively. While mean AODs of GEOS-Chem and HSRL-2 are similar, the spread of the GEOS-Chem AOD is greater than the AOD

spread of HSRL-2 (Figure 5). The *FB*s of WRF-FINN and EAM-E3SM are -0.13 and -0.12, respectively. However, the *FE*s

of both models are about a factor of two greater than the magnitude of their *FB*s, suggesting that similarities in the mean AODs are the result of cancellation of high and low AOD biases. The FT AOD comparisons between the models and the HSRL-2 reveal similar *FE* and *FB* relationships (Figure S18). These findings are consistent with the AOD comparisons between the models and aircraft during ORACLES 2016 in Shinozuka et al. (2020). Doherty et al. (2022) noted that extinction profiles of WRF-CAM5 and GEOS-FP generally tend to be lower than those measured by the HSRL-2. They also found that the extinction

profiles are more vertically diffuse with weaker vertical gradients than the lidar measurements. Evaluation of modeled AODs against those from the 4STAR are presented in Figure 11. All models except for GEOS-FP and MERRA-2 show significantly different statistical results with respect to the HSRL-2 and the 4STAR. AODs from the HSRL-2 were generally obtained further south than from the 4STAR. The statistical differences in comparing with the two instruments are due to the different sampling locations and times.


Scatterplots for total-column AOD comparisons between the models and the HSRL-2 for August 2017 are shown in Figure 12. AODs in WRF-CAM5, GEOS-FP, and MERRA-2 are mostly lower than AODs in HSRL-2, showing FBs of -0.31, -0.28, and -0.16, respectively. AODs in WRF-FINN during August 2017 show a factor of five higher in *FE* than the magnitude of *FB*. The slightly negative *FB* of -0.06 is consistent with over half WRF-FINN samples having lower AODs than the AODs

from the HSRL-2. AODs in GEOS-Chem are biased high, with an *FE* and *FB* of 0.28 and 0.17, respectively. AOD comparisons between the models and the 4STAR reveal that the models underestimate AODs for measured AODs that are above 1 (Figure 13). In general, the models underestimate AODs for measured AODs that are above 0.8, an indication that models are unable to reproduce high AODs. Table 2 summarizes the statistics for the scatter plots between Figure 10 and Figure 13.

## 4    Discussion on model deficiencies for future investigations

ESMs are complex and nonlinear systems, so AOD errors are likely caused by numerous factors. Identifying the exact causes of AOD biases is challenging and entails a detailed examination of model source codes. Here, we present aspects of the models that may explain their biases in simulated AOD relative to those measured by airborne lidar, which establishes a starting point for a future in-depth investigation. The assimilation of clear-sky MODIS AODs in the two assimilation systems (i.e., GEOS-FP and MERRA-2) may explain their better performance compared to other models in simulating AODs, especially in August

2017. Despite a lack of MODIS clear-sky AOD retrievals over regions with expansive cloud presence, such as in the austral spring of the SE Atlantic, AOD assimilation is still beneficial for minimizing AOD errors in ESMs. The mean and median AOD and the AOD fraction in the FT in WRF-FINN generally agree well with those from aircraft measurements. WRF-FINN is also the only model in this study that includes a plume rise parameterization. The importance of the inclusion of a plume rise model for simulating high AODs in this region is unclear since fire emissions in southern Africa already take place at

elevated altitudes. Nonetheless, the smoke top heights in the remaining models generally agree with those from lidar measurements (Shinozuka et al., 2020).

The rate of primary organic aerosol (POA) removal and the secondary organic aerosol (SOA) production influences the simulated AOD (Hodzic et al., 2020). For example, the negligible production of SOA in WRF-CAM5, GEOS-FP, MERRA-
2, and ALADIN may be contributing to a low bias in simulated AOD. For GEOS-Chem, GEOS-FP and MERRA-2, their aerosol optical properties are assumed to be fixed and do not account for particle evolution during transport. Even though the production of SOA is introduced in the other models, the assumed processes may be oversimplified such that its production is based on precursors at a fixed time-scale without a detailed consideration for chemistry. Moreover, these models do not treat photochemical loss of SOA as shown by its excessive OC according to Shinozuka et al. (2020). Errors in the treatment of
aerosol hygroscopicity may also play a crucial role in the aerosol evolution and subsequent AOD biases. Although the AOD fraction in the FT in WRF-CAM5 has a good agreement with lidar measurements, Shinozuka (2020) found that the PBL height of this model was a few hundred meters higher than that in lidar cloud-top measurements in September 2016, possibly leading to overactive entrainment and aerosol removal. While the selection of emission inventory alone impacts simulated AODs (Pan et al. 2020), the use of monthly emission inventory in both EAM-E3SM and ALADIN instead of diurnally-varied emissions
as in other models could further be responsible for some of the errors. These deficiencies suggest that AOD errors in each model are likely driven by multiple factors, and a more in-depth model-specific analysis would be needed to investigate model deficiencies that leverages multiple degrees of freedom.

## 5    Summary and Conclusions

The AOD fraction in the FT, full-column AODs, and FT AODs from Earth system models were examined over the southeast
Atlantic Ocean during the September 2016 and August 2017 time frame of the NASA ORACLES field campaign. The modeled AODs were compared against each other and then evaluated against aircraft-based measurements, and as such, were spatially and temporally interpolated to the locations of the HSRL-2 and the 4STAR aircraft-based measurements. To account for the presence of decoupled PBLs in the models, the level of maximum vertical gradient of specific humidity profiles from each model was used to derive PBL heights.


Over most of the southeast Atlantic, more than half of the total column AOD from MERRA-2, GEOS-Chem, and EAM-E3SM resides in the FT. ALADIN shows over half the columnar AOD in the FT primarily north of 20°S. WRF-CAM5, MERRA-2, and GEOS-FP show high fractions of AOD in the FT off coastal Namibia and Angola, but the FT fraction markedly decreases northwestward from the coast. The proportion of AOD in the FT compared to the total-column AOD ranges between 44% and
74% in September 2016 across seven models within the region bounded by 25°S – 0° and 15°W – 15°E (excluding land). During August 2017, the range is between 54% and 71% across five models and the spread of the fraction in each model is

smaller than the individual model spread in September 2016. ALADIN and GEOS-Chem show similar in aerosol plume patterns when compared to the above-cloud aerosol product from MODIS during September 2016, but none of the models show a similar above-cloud plume patterns as MODIS does in August 2017. The HSRL-2 clear-sky AOD measurements from September 2016 are used to infer observational-based fractional AOD in the FT by using modeled PBL heights to separate the FT and PBL. Results indicate that WRF-CAM5 separates AOD fraction between the FT and the PBL reasonable well, but its AOD tends to be lower than aircraft-based measurements. AOD fractions in the FT for GEOS-Chem and EAM-E3SM are, respectively, 10% and 15%, lower than the AOD fractions from the HSRL-2. While both models generate similar mean AOD as those from the HSRL-2, their similarities are the result of cancellation of high and low AOD biases. GEOS-FP, MERRA-2, and ALADIN produce less aerosol loading and tend to misplace more aerosols in the PBL compared to HSRL-2 measurements. The model evaluation during ORACLES 2017 shows that the models generally underestimate AODs when measured AODs exceed 0.8, indicating their limitations at reproducing high AODs.

The modeling differences in the column AOD, FT AOD, and the vertical apportioning of AOD in this study emphasize the need to continue improving the accuracy of AOD and PBL height distributions. These differences affect the sign and magnitude of the net aerosol radiative forcing, especially when aerosols are in contact with different cloud phases (i.e., low- and mid-level clouds). In conditions where aerosols are in both the FT and in contact with clouds, both the aerosol direct and indirect forcing are significant. Aerosol direct and semi-direct forcing usually play a larger role for free tropospheric aerosols; however, both types of forcing could compete with the aerosol indirect forcing when aerosols are in contact with clouds in the FT.

**Data availability**

The P3 and ER2 data (ORACLES Science Team, 2020a, b, c) are available through https://doi.org/10.5067/Suborbital/ORACLES/ER2/2016_V3 , https://doi.org/10.5067/Suborbital/ORACLES/P3/2016_V3, https://doi.org/10.5067/Suborbital/ORACLES/P3/2017_V3.

**Author contributions**

CJF, SPB, RAF, SEL, MSK, MSR, KP and YS operated instruments during the ORACLES intensive observation periods. GAF, PC, AMdS, PES, CH, ZX, MM, KML, RG, QW, YC, YF, and SAC delivered model products. KGM delivered the MODIS above-cloud aerosol product. LP and JMR developed the methodology of determining MBL height. IC and JR formulated the model–observation comparison. IC and LG organized all products, performed statistical analyses and visualized the results. IC wrote the first draft. IC, LG, CJF, YS, SJD, MSD, GAF, GRC, PC, AMdS, PES, CH, ZX MM, KML, QW, YF, SPB, RAF, SEL, MSK, KP, MSR, JMR, LP, AAA, RW, PZ, SAC, and JR edited the manuscript. RW, PZ, and JR led the efforts to acquire funding for the ORACLES mission and co-led the five year investigation.

**Competing interests**

Paquita Zuidema is a guest editor for the ACP Special Issue: "ACP special issue: New observations and related modelling studies of the aerosol–cloud–climate system in the Southeast Atlantic and southern Africa regions" The rest of authors declare
that they have no conflict of interest.

**Special issue statement**

This article is part of the special issue "New observations and related modelling studies of the aerosol-cloud-climate system in the Southeast Atlantic and southern Africa regions (ACP/AMT inter-journal SI)". It is not associated with a conference.

**Acknowledgements**

Part of the computation in this paper was performed at the Supercomputing Center for Education & Research (OSCER) at the University of Oklahoma (OU). We acknowledge use of the WRF-Chem preprocessor tool (mozbc, fire_emiss) provided by the Atmospheric Chemistry Observations and Modeling Lab (ACOM) of NCAR. YF would like to acknowledge the support provided by the U.S. DOE Office of Science, under Contract No. DE-AC02-06CH11357, and thank the Energy Exascale Earth System Model (E3SM) project funded by the DOE Office of Biological and Environmental Research and all the E3SM project
team members for their efforts in developing and supporting the E3SM.

**Financial support**

ORACLES was funded by NASA Earth Venture Suborbital-2 (NNH13ZDA001N-EVS2).

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

**Tables**

Table 1. General specifications including emission, transport, and deposition processes of the models in this study. The acronyms of GFED, FINN, QFED, stand for Global Fire Emission Database, Fire INventory from NCAR, and Quick Fire Emissions Dataset, respectively.

| Model name | WRF-CAM5 | WRF-FINN | GEOS-FP | GEOS-Chem | MERRA-2 | EAM-E3SM | ALADIN-Climate |
|---|---|---|---|---|---|---|---|
| Model version | | 4.2.2 | 5.13.1 (2016) and 5.16 (2017) | 12.0.0 | 5.12.4 | V1 | |
| Domain extent | 41°S-14°N, 34°W-51°E | 37°S-24°N, 31°W-51°E | Global | Global | Global | Global | 37°S-9°N; 33°W-45°E |
| Horizontal grid resolution (lon × lat or km) | 36 km | 36 km | 0.3125°×0.25° (2016) 0.3125°×0.125° (2017) | 2.5° × 2° | 0.625° × 0.5 | 110 km | 12 km |
| Vertical levels | 75 | 73 | 72 | 72 | 72 | 72 | 91 |
| Initializing meteorology | NCEP Final Analysis | ERA5 | GEOS-FP | GEOS-FP | MERRA-2 | ERA-INT | ERA-INT |
| Initialization frequency | 5-day | 5-day | Daily | Hourly | Daily | 6-hourly | Once (at the beginning) |
| Cloud scheme | 2-moment microphysics, 1-moment macrophysics | 2-moment microphysics | 1-moment scheme (Bacmeister et al., 2006; Moorthi and Suarez, 1992) | Same as GEOS-FP | (Same as GEOS-FP) | Updated 2 moment microphysical scheme, version 2 of Morrison and Gettelman (2008) (Gettelman et al., 2015) | 1-moment scheme (Ricard and Royer, 1993) |
| PBL scheme | Bretherton and Park (2009) | Mellor-Yamada by Janjić (1990, 1994) | Lock et al. (2000) based on the Bulk Richardson number scheme of Louis and Geleyn (1982) | VDIFF: non-local scheme formulated by Holtslag and Boville (1993) | (Same as GEOS-FP) | CLUBB by Larson and Golaz (2005) | Cuxart et al. (2000) |
| Aerosol scheme | MAM3 | MOZART-MOSAIC | GOCART | GEOS-Chem standard | GOCART | MAM4 | TACTIC scheme |
| Aerosol assimilation/ radiation/ cloud microphysics | No/Yes/Yes | No/Yes/Yes | Yes/Yes/Yes | No/Yes/Yes | Yes/Yes/Yes | No/Yes/Yes | No/Yes/No |
| Emission | QFED2 are provided daily and added hourly using a fixed diurnal profile at surface level. A diurnal cycle is imposed to match observed behavior. Particle emissions are at a fixed size distribution, | FINNv2 with daily temporal resolution is applied with the WRAP(Western Regional Air Partnership) emission profile to allocate the emissions to a diurnal cycle. The emissions are distributed vertically in | QFED2 with daily resolution, emitted to levels within the PBL. No plume rise is used. | QFED2 with daily resolution applied in the model does not contain any diurnal cycle. It is assumed that 65% of QFED2 emission is uniformly distributed within PBL while the rest is uniformly distributed | (Same as GEOS-FP) | GFED4 monthly emissions are used for primary OC and BC from biomass burning. Following the MAM4 in Liu et al. (2016), biomass burning BC and OC are first emitted to a fresh carbon aerosol | GFED4 with monthly resolution applied in the model that does not contain any diurnal cycle. Emissions are only applied at the surface without a plume rise model. The biomass |

| | | | | | | | |
|---|---|---|---|---|---|---|---|
| | after which they evolve freely. | different levels using the plume rise model. | | in the free troposphere up to 5.5 km. Particle emissions are at a fixed size distribution, mostly in the accumulation mode. AOD is calculated assuming lognormal size distributions of externally mixed aerosols and accounts for hygroscopic growth. | | size mode and then grow into the accumulation size mode in aging, due to the coating of soluble materials. All the SOA formation is in the accumulation mode. | burning aerosol emissions have been scaled up by a factor of 1.5 for BC and OC. Particle emissions are at a fixed size distribution, after which they evolve freely. |
| **Transport** | After the point of emission, all chemical tracers and aerosols are fully coupled with the radiation, chemical, and aerosol microphysics schemes. Model meteorology is reinitialized from reanalysis every 5 days and otherwise evolves freely. Aerosols are copied over from the previous 5-day run cycle. | Model meteorology is reinitialized from ERA5 reanalysis data every 5 days and chemistry fields by the end of every 5 days are used in initializing the next simulation. | Model is driven by GEOS-FP meteorology that assimilates conventional near-real time satellite and sub-orbital meteorological observations. | The model is driven by assimilated meteorological data GEOS-FP from NASA GMAO. | Model is driven by reanalyzed meteorology. | The model was ran in the nudged mode. ERA-INT reanalysis meteorological data for year 2016 was used. | After the point of emission, all aerosol types are fully coupled with the radiation but not with aerosol microphysics schemes. The model lateral boundary is driven by the ERA-Interim reanalysis. Spectral nudging is applied to wind, surface pressure, temperature and specific humidity, using a constant rate above 700 hPa and a relaxation zone between 700 and 850 hPa.Sea surface temperatures are prescribed. |
| **Deposition** | Aerosols and gases are both subject to wet and dry deposition. Preliminary analysis suggests that parameterized convective deposition is small compared to deposition through the microphysics scheme | Aerosols and gases are both subject to wet and dry deposition. Dry deposition follows Wesely (2007), which models deposition as a series of resistors. Wet deposition includes the removal through | Aerosols and gases are both subject to wet and dry deposition, including gravitational settling, large-scale wet removal, and convective scavenging. | Aerosols and gases are both subject to wet and dry deposition. Dry deposition follows the standard resistance-in-series scheme, accounting for turbulent transfer and gravitational settling. | (Same as GEOS-FP) | Dry and wet deposition of gas and aerosol species are treated in the model as described in Liu et al. (2016) and Wang et al. (2020). | Aerosols and gases are both subject to wet and dry deposition. Dry deposition is adapted from Reddy et al. (2005). Wet deposition in-cloud is based on Giorgi and |

| | | | |
|---|---|---|---|
| (i.e., cloud droplet activation), and that total wet deposition over the stratocumulus region might be underestimated. | convective and grid-scale precipitation. | Wet deposition accounts for scavenging in both convective updrafts and large-scale precipitation and distinguishes ice/snow scavenging from rain scavenging. | Chameides (1986) and below-cloud scavenging by Morcrette et al. (2009). |

Table 2. A statistical summary of the scatter plots between Figure 10 and Figure 13. The sample size of each comparison (*N*) is indicated in the parenthesis.

| | Fitting | $R_S$ | RMSE | FE | FB |
|---|---|---|---|---|---|
| ORACLES 2016: Model vs HSRL-2 (*N*=79) | | | | | |
| WRF-CAM5 | 0.57X + 0.01 | 0.90 | 0.05 | 0.57 | − 0.54 |
| WRF-FINN | 1.10X − 0.02 | 0.94 | 0.12 | 0.31 | − 0.13 |
| GEOS-FP | 0.70X + 0.00 | 0.89 | 0.06 | 0.40 | − 0.38 |
| GEOS-Chem | 1.32X − 0.03 | 0.92 | 0.11 | 0.36 | 0.00 |
| MERRA-2 | 0.78X − 0.01 | 0.90 | 0.06 | 0.40 | − 0.39 |
| EAM-E3SM | 0.73X +0.05 | 0.90 | 0.08 | 0.21 | − 0.12 |
| ALADIN | 0.55X + 0.02 | 0.79 | 0.07 | 0.70 | − 0.60 |
| ORACLES 2016: Model vs 4STAR (*N*=90) | | | | | |
| WRF-CAM5 | 0.79X + 0.06 | 0.79 | 0.07 | 0.24 | − 0.01 |
| WRF-FINN | 0.61X + 0.12 | 0.56 | 0.12 | 0.36 | − 0.02 |
| GEOS-FP | 0.73X + 0.00 | 0.77 | 0.08 | 0.41 | − 0.35 |
| GEOS-Chem | 1.72X − 0.18 | 0.75 | 0.20 | 0.46 | − 0.04 |
| MERRA-2 | 0.72X + 0.02 | 0.73 | 0.08 | 0.35 | − 0.25 |
| EAM-E3SM | 0.61X + 0.14 | 0.65 | 0.10 | 0.31 | + 0.07 |
| ALADIN | 0.48X + 0.19 | 0.45 | 0.12 | 0.35 | + 0.10 |
| ORACLES 2017: Model vs HSRL-2 (*N*=69) | | | | | |
| WRF-CAM5 | 0.34X + 0.14 | 0.75 | 0.05 | 0.37 | − 0.01 |
| WRF-FINN | 0.59X + 0.17 | 0.56 | 0.13 | 0.33 | − 0.06 |
| GEOS-FP | 0.65X + 0.04 | 0.88 | 0.05 | 0.29 | − 0.28 |
| GEOS-Chem | 0.80X + 0.14 | 0.68 | 0.12 | 0.28 | + 0.17 |
| MERRA-2 | 0.67X + 0.07 | 0.92 | 0.06 | 0.19 | − 0.16 |
| ORACLES 2017: Model vs 4STAR (*N*=62) | | | | | |
| WRF-CAM5 | 0.19X + 0.31 | 0.49 | 0.12 | 0.35 | + 0.02 |
| WRF-FINN | 0.77X + 0.25 | 0.68 | 0.17 | 0.41 | + 0.34 |
| GEOS-FP | 0.54X + 0.14 | 0.83 | 0.08 | 0.22 | − 0.06 |
| GEOS-Chem | 0.51X + 0.40 | 0.55 | 0.20 | 0.47 | + 0.42 |
| MERRA-2 | 0.45X + 0.19 | 0.85 | 0.08 | 0.19 | − 0.01 |

**Figures**

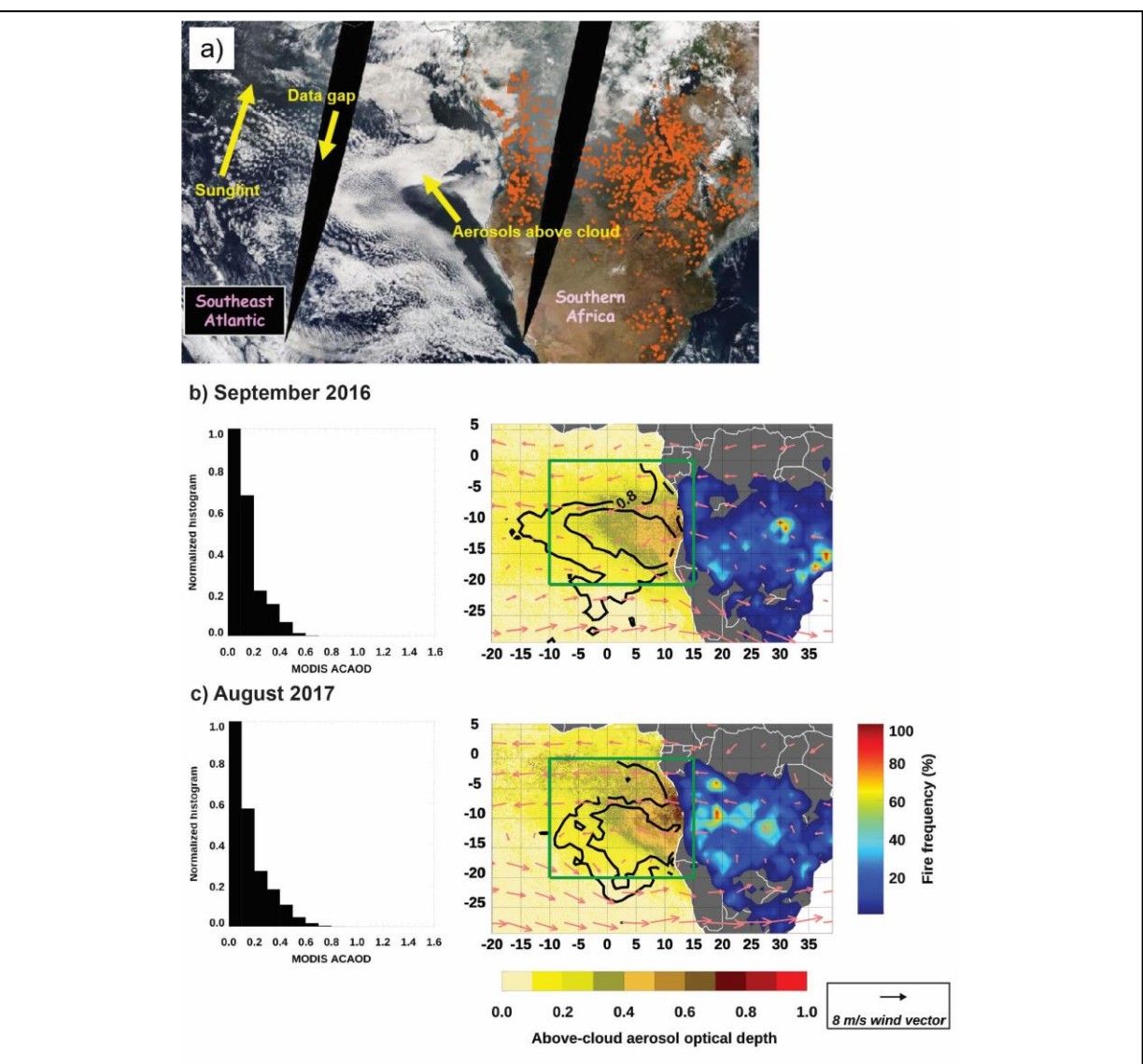

Figure 1. The MODIS Terra true color image with fire locations (in orange) over the southeast Atlantic and southern Africa on 12 August 2017 (a). Monthly mean oceanic ACAOD from MODIS based on the Meyer et al. (2015) above-cloud aerosol algorithm, fire frequency (detection confidence above 70%), and maritime low-level (clouds with tops up to 2.5 km altitude) cloud fractions (0.8 and 0.9) accompanied by normalized histograms of the satellite ACAOD from the regions delineated by green boxes (excluding the land) for (b) September 2016 and (c) August 2017. Pink arrows represent 600 mb wind vectors from the National Centers for Environmental Prediction (NCEP) Reanalysis data set. The satellite image is adapted from NASA EOSDIS Worldview (https://worldview.earthdata.nasa.gov/).

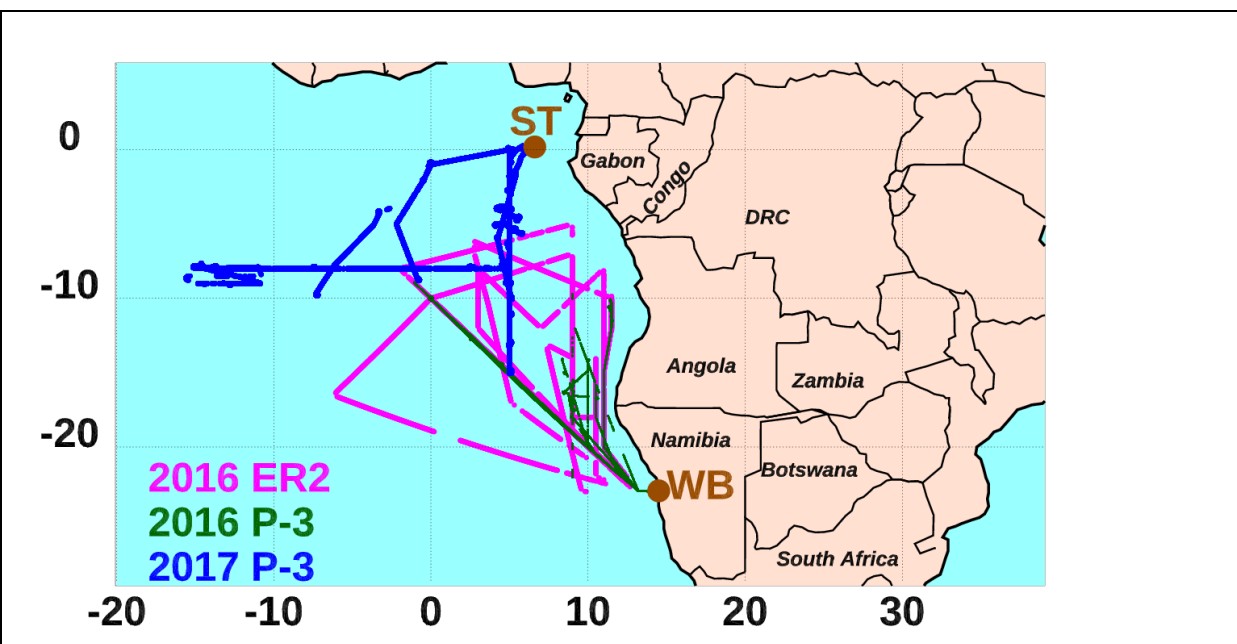

Figure 2. Locations of AOD measurements from the HSRL-2 aboard the ER-2 during ORACLES 2016 (in magenta), 4STAR aboard the P-3 during ORACLES 2016 (in green) and both instruments aboard the P-3 during ORACLES 2017 (in blue). Walvis Bay and São Tomé are denoted by WB and ST, respectively.

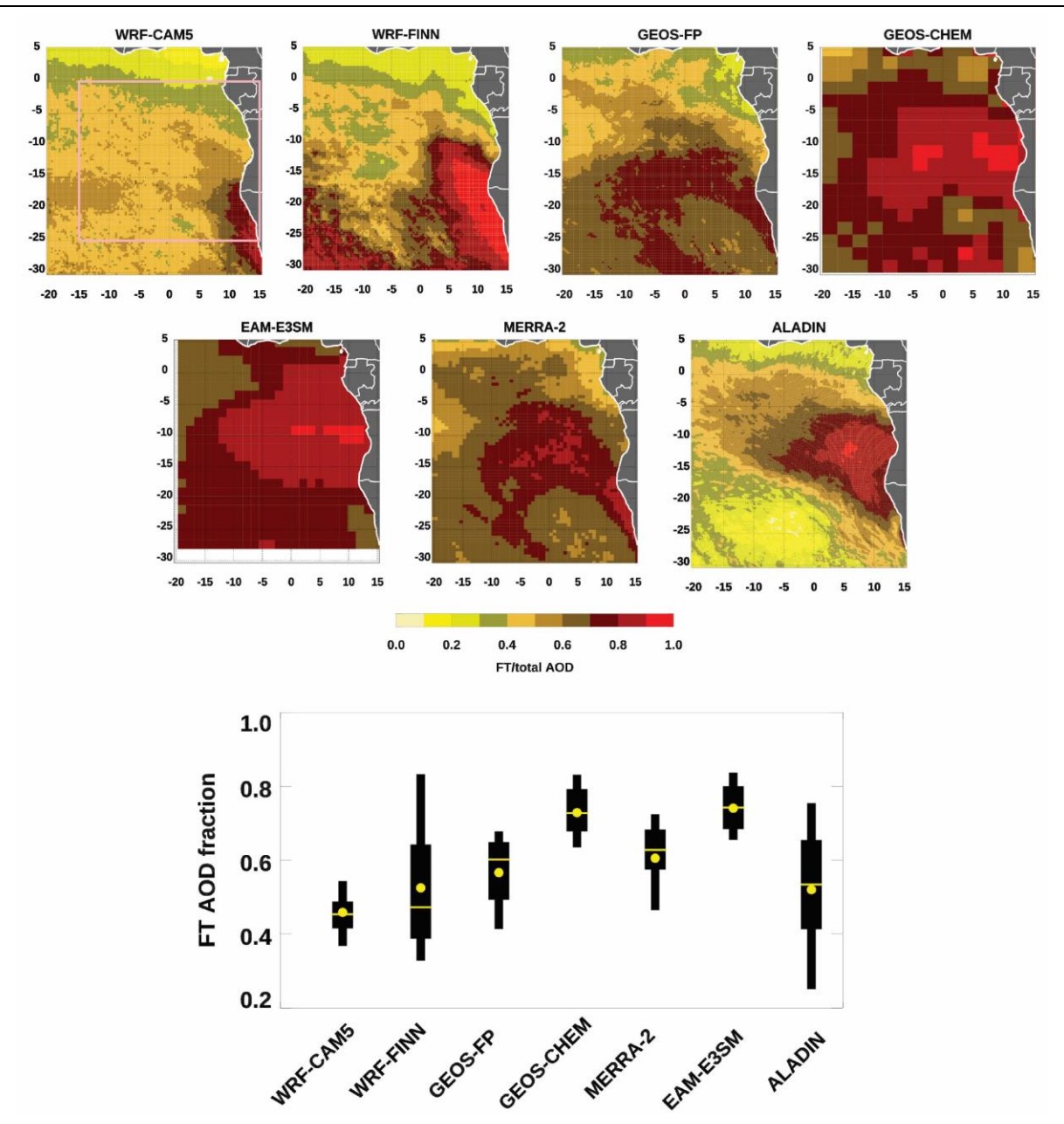

Figure 3. Fractions of free tropospheric AOD to total-column AOD (at 550 nm) during September 2016 over the southeast Atlantic. A ratio of 1 means that the total AOD contribution is in the FT. The pink box represents the boundary (25°S – 0° and 15°W – 15°E) of the region used for the results in the box-whisker plot at the bottom of the figure. The box-whisker plot summarizes the 10[th] (whisker), 25[th] (box), 50[th] (yellow horizontal line), 75[th] (box), and 90[th] (whisker) percentiles of the ratios of AOD fractions in the FT for the ESMs bounded by the region in the pink box. The yellow dots represent the mean ratio of each scenario.

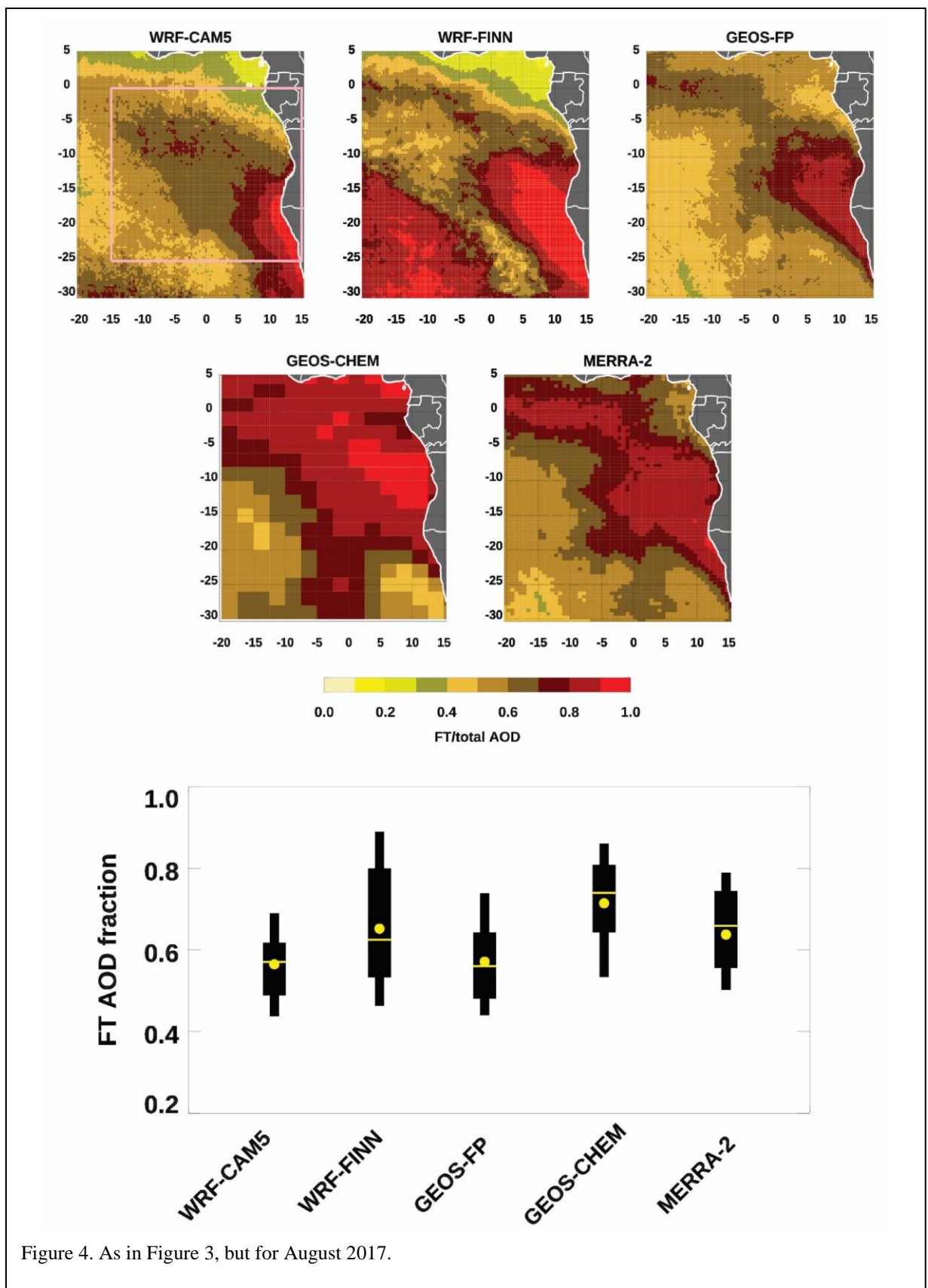

Figure 4. As in Figure 3, but for August 2017.

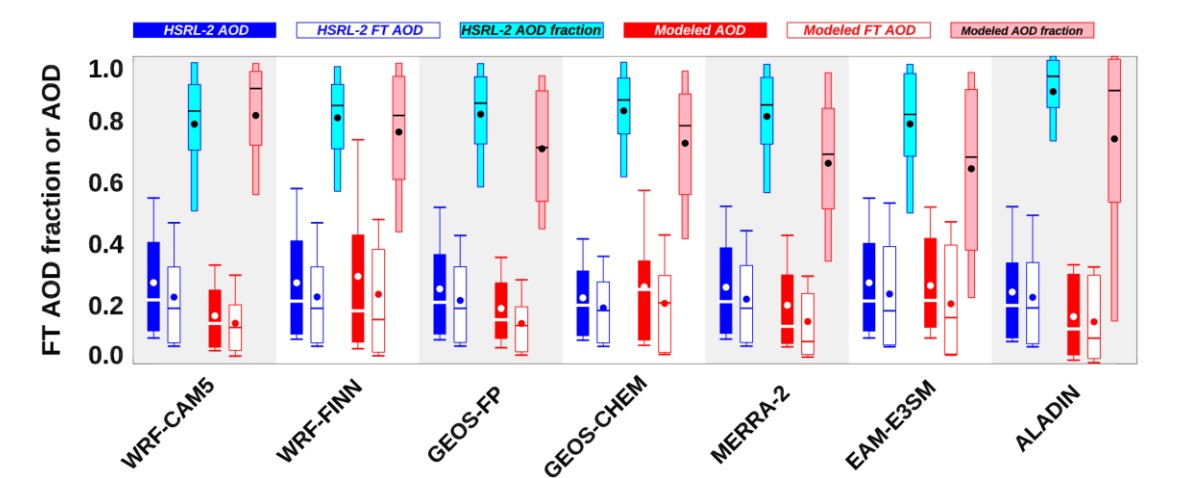

Figure 5. Comparisons of AOD, FT AOD, and AOD fraction in the FT between those from the HSRL-2 and the models during September 2016. As indicated by the legend at the top, HSRL-2 quantities of AOD and FT AOD are displayed in blue and AOD fraction in the FT are displayed in cyan. Analogous modeled quantities are shown in red and pink. The box-whiskers show the 10th (whisker), 25th (box), 50th (horizontal lines), 75th (box), and 90th (whisker) percentiles and the dots represent the means. A total of 1,334 matchups between the HSRL-2 and each model are used. For clarity, the alternating gray and white background separates each set of box-whisker from a model to another.

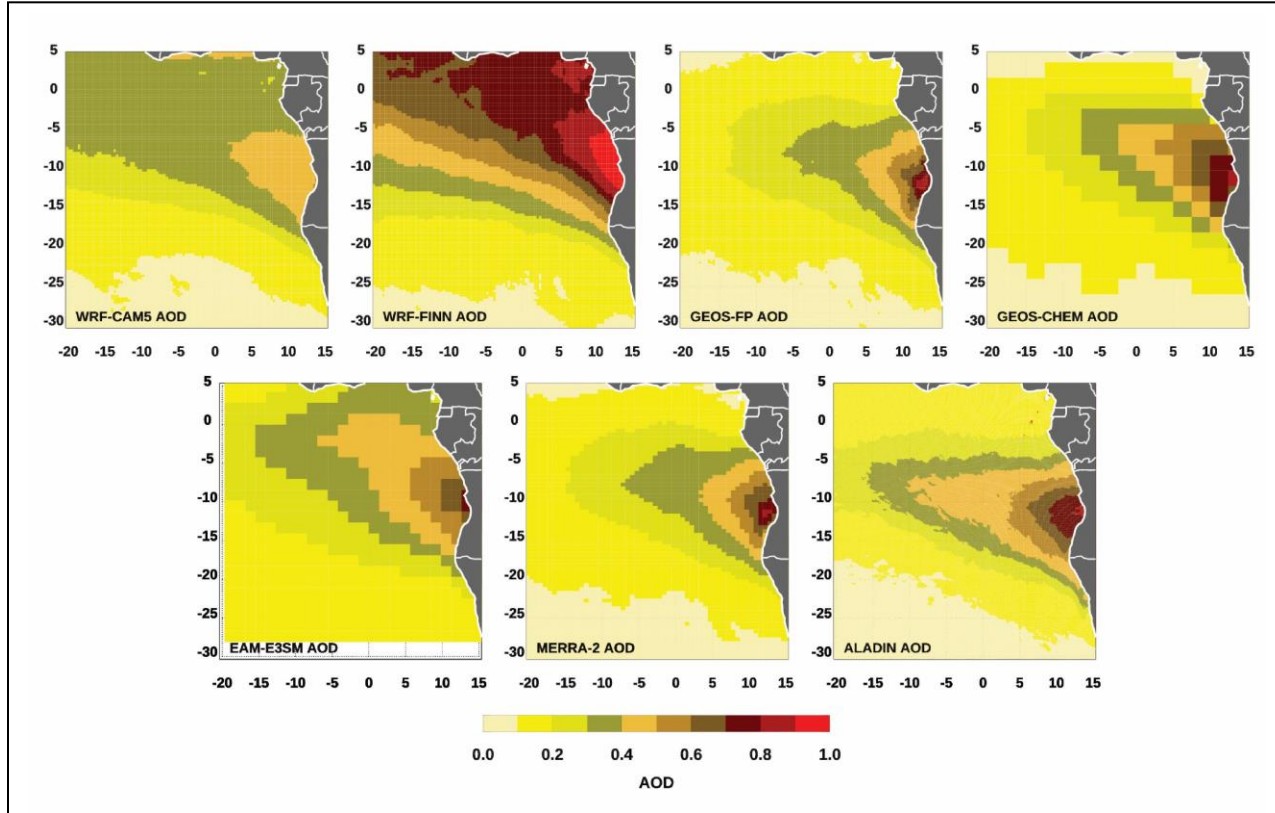

Figure 6. Monthly mean modeled total-column AOD at 12 UTC for September 2016.

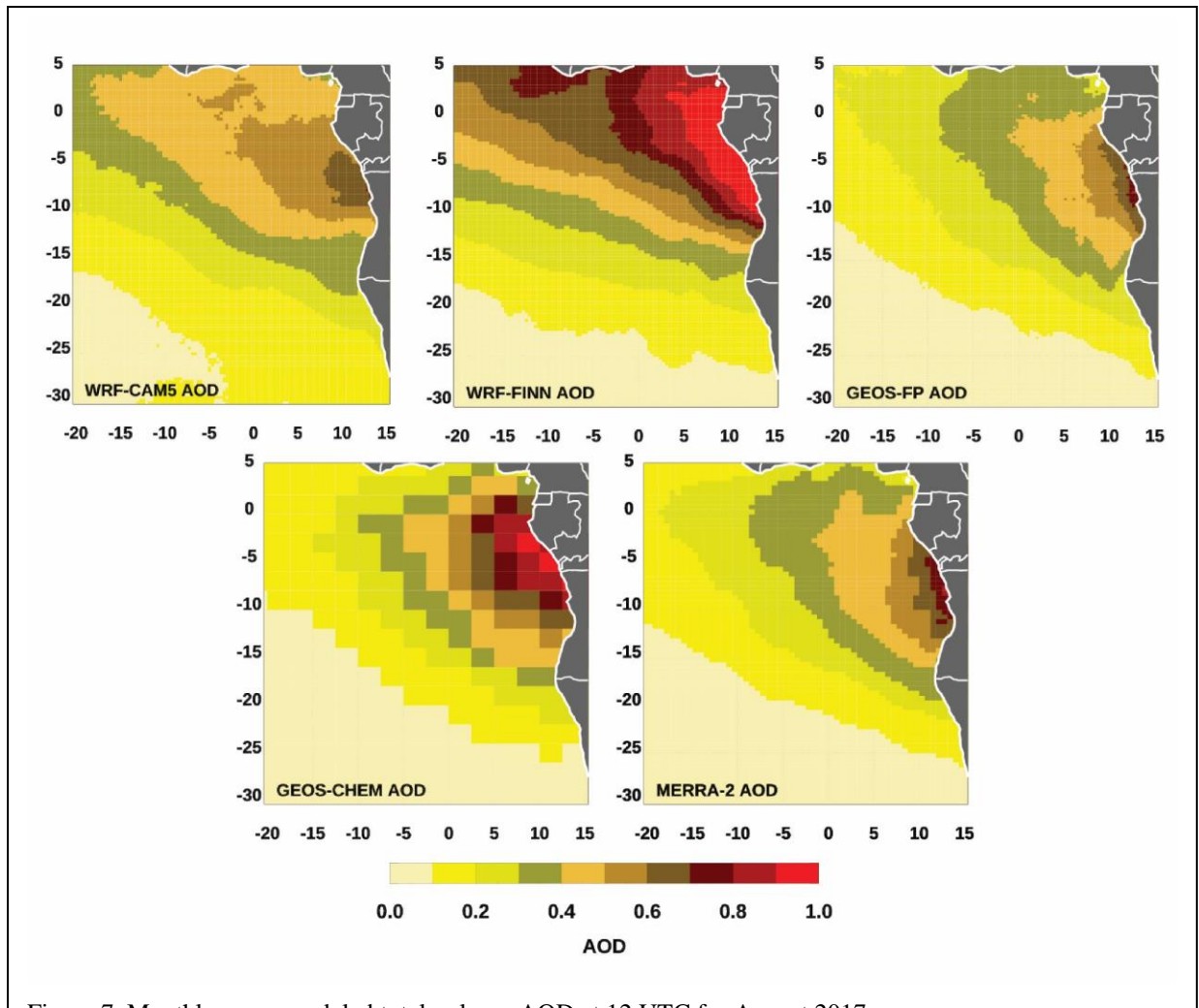

Figure 7. Monthly mean modeled total-column AOD at 12 UTC for August 2017.

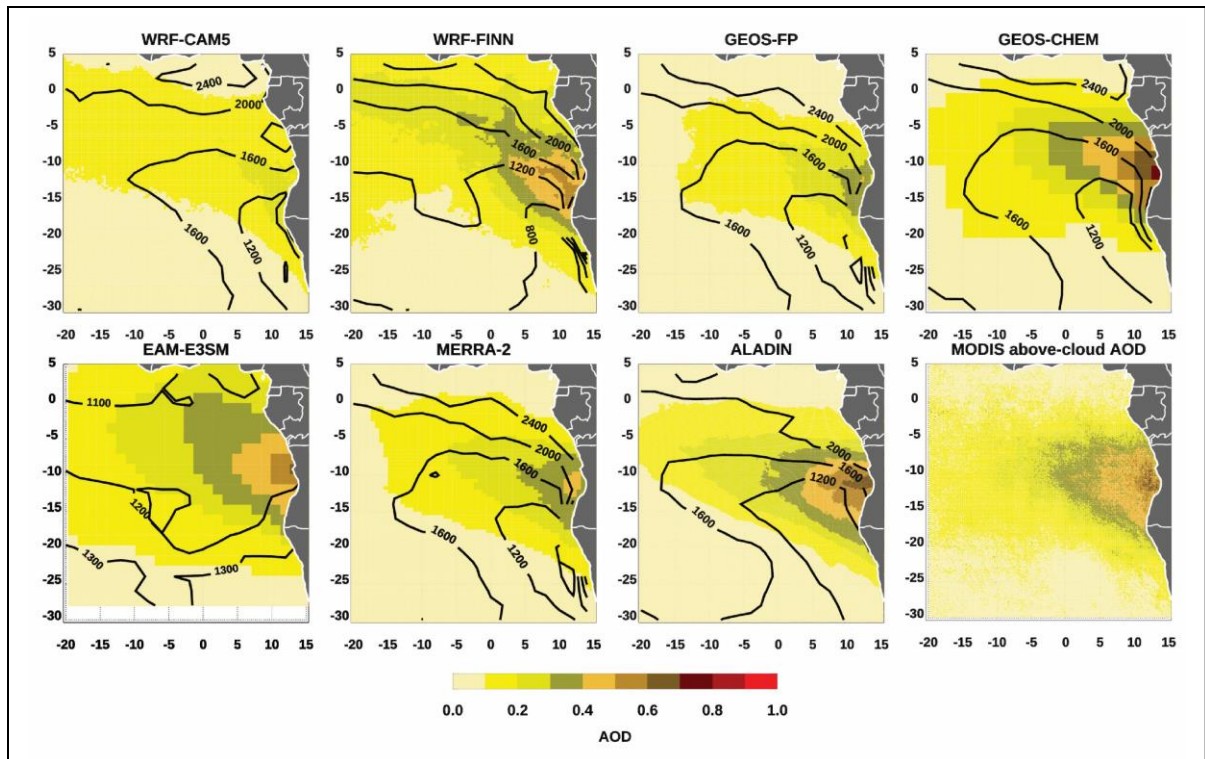

Figure 8. Monthly mean modeled AOD in the FT at 12Z and MODIS (Terra and Aqua) ACAOD for September 2016. Contours are PBL heights in meters.

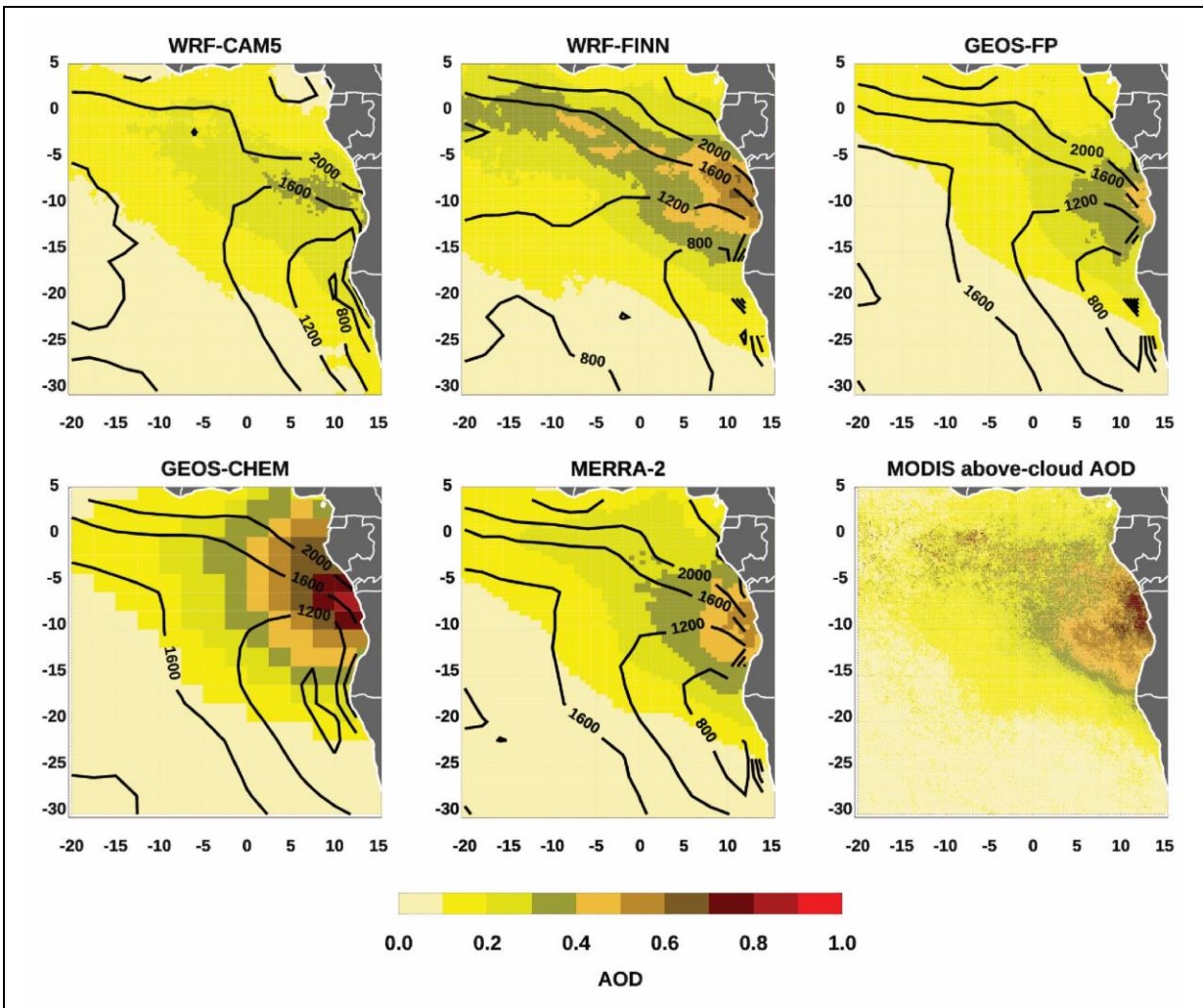

Figure 9. Monthly mean modeled AOD in the FT at 12Z and MODIS (Terra and Aqua) above-cloud AOD for August 2017. Contours are PBL heights in meters.

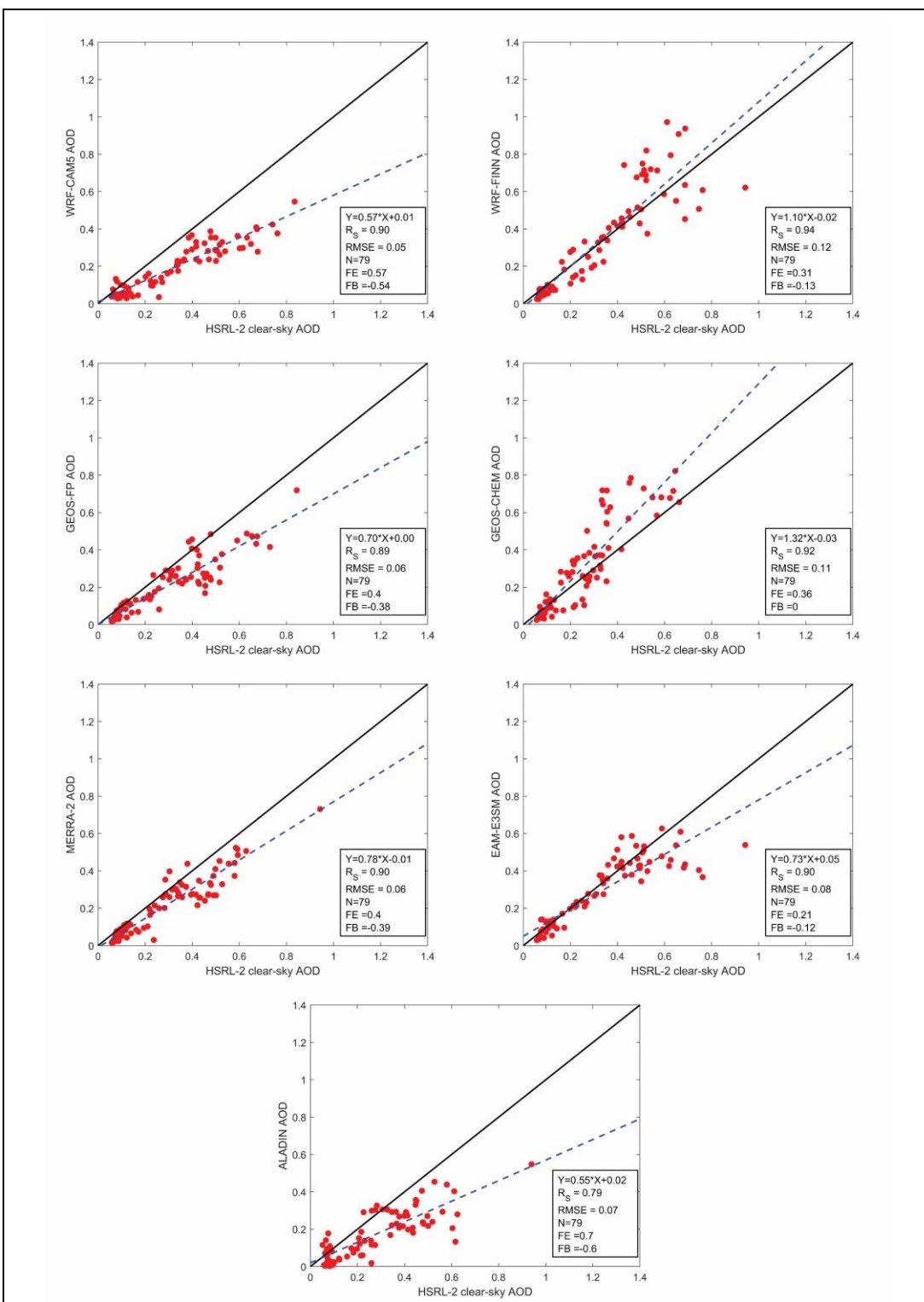

Figure 10. Scatter plots comparing full-column AOD (at 550 nm) from the models to HSRL-2 clear-sky AOD during the September 2016 deployment of the ORACLES field experiment. An ordinary least square (dashed blue lines) is used to estimate the linear fit.

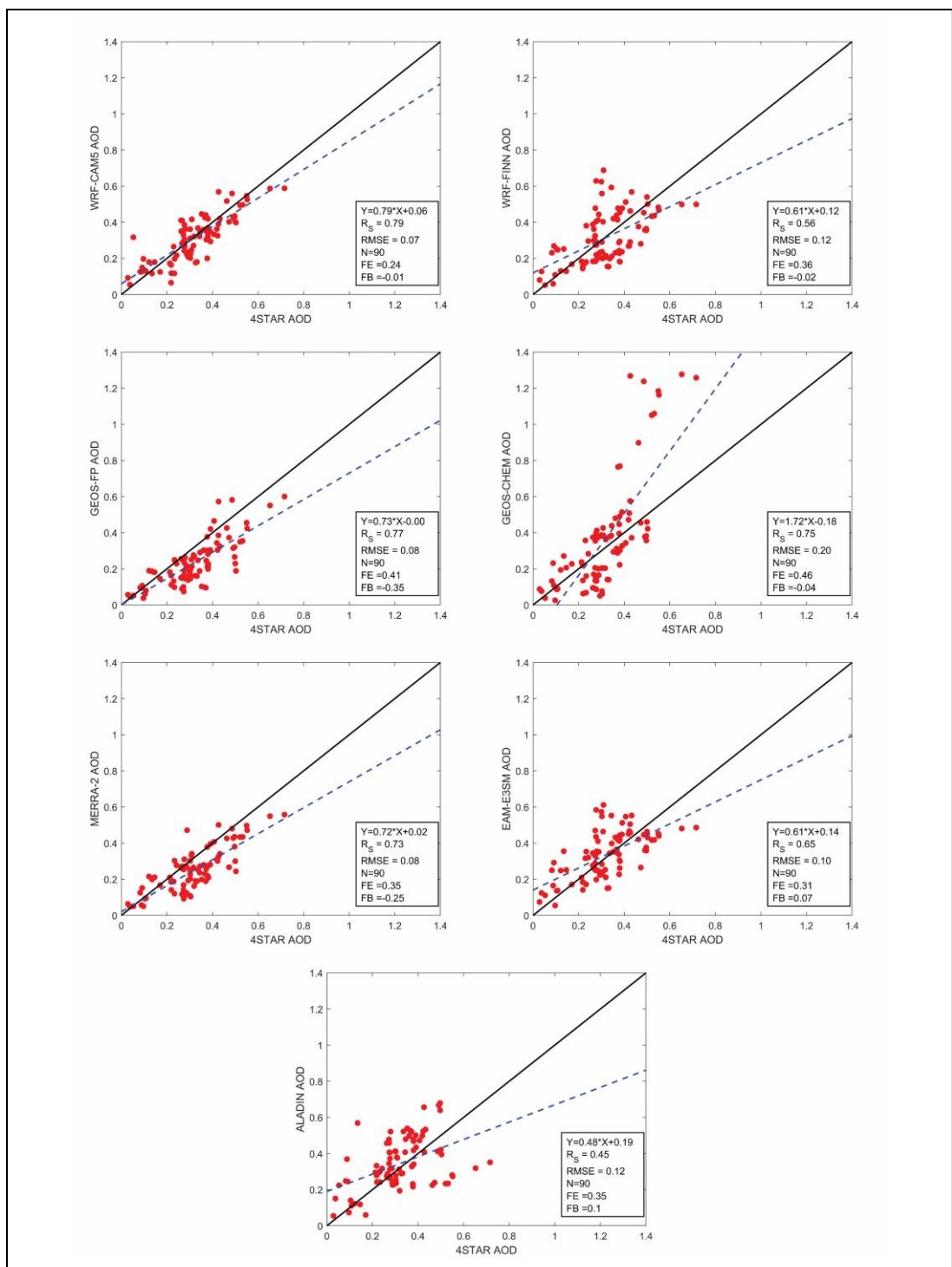

Figure 11. Scatter plots comparing modeled AOD (at 550 nm) and 4STAR AOD during September 2016 of the ORACLES field experiment.

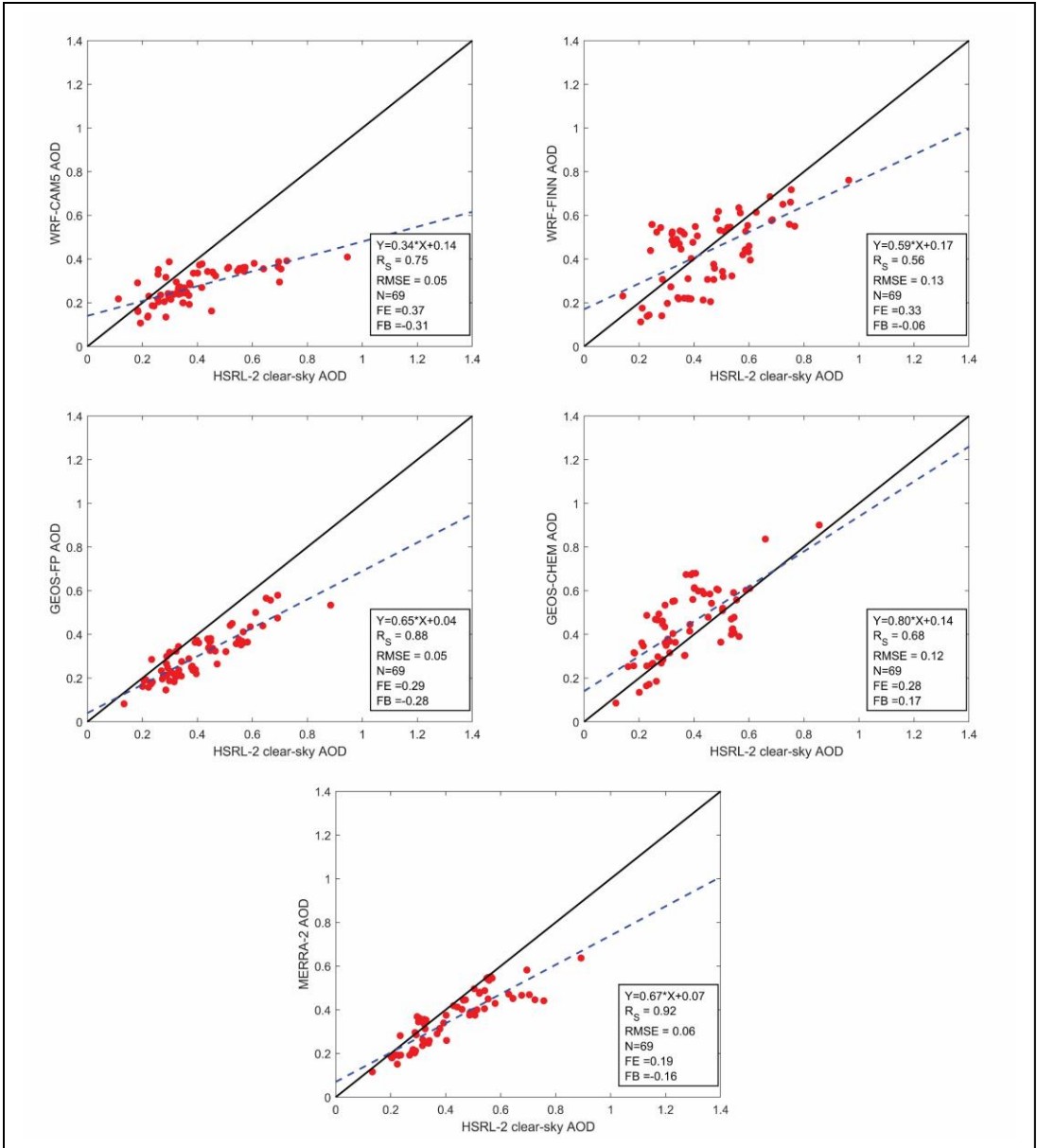

Figure 12. Scatter plots comparing full-column AOD (at 550 nm) among models and HSRL-2 clear-sky AOD during August 2017 of the ORACLES field experiment.

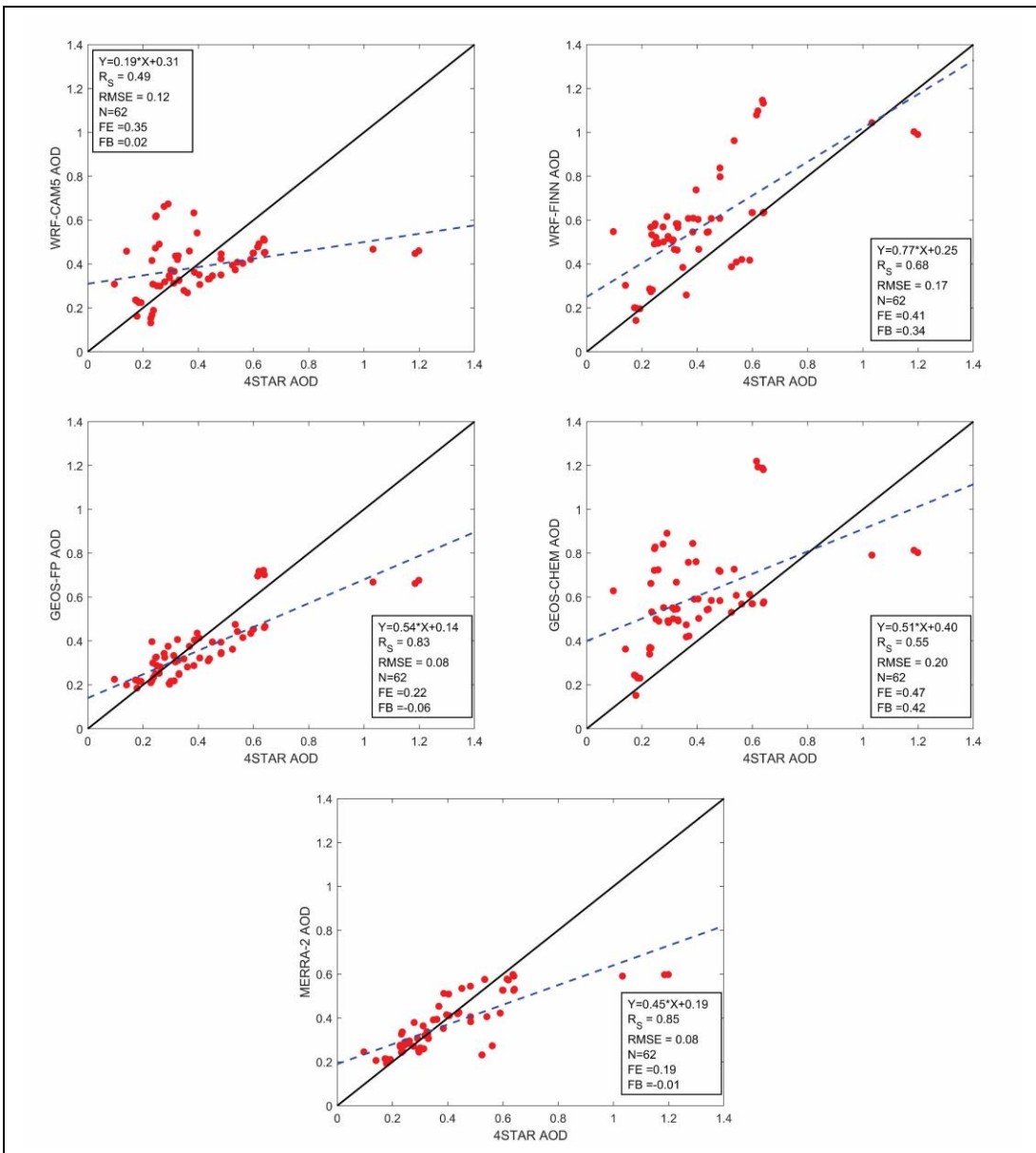

Figure 13. Scatter plots comparing modeled AOD (at 550 nm) and 4STAR AOD during August 2017 of the ORACLES field experiment.