# Peer review of "On the differences in the vertical apportionment of modeled aerosol optical depth over the southeast Atlantic"

_Atmospheric Chemistry and Physics, 2022_

## Author Comment (AC1)

**Response to reviewers for Chang et al. (acp-2022-496)**

We thank the reviewers for their constructive comments and suggestions. We are glad to hear that the reviewers find this work well-written and suitable for consideration to be published in ACP. Our responses are in bold-free texts and the reviewers' comments are in bold texts.

**Reviewer 1**

**General comments**

**This is the first review of the article by Chang et al. "On the differences in the vertical distribution of modeled aerosol optical depth over the southeast Atlantic". A comparative study of aerosol optical thickness calculated by seven models and measured by various in-situ (airborne) instrumentation is documented in the manuscript.**

**The manuscript is well written and lets you read it well. So apart from a few stylistic details or minor technical corrections, the exposition is adequate for publication.**

**Scientifically, the chosen statistical tools are also adequate, although I personally would prefer to read predictive statements about the significance of the differences found between models and measurements, even in the presence of a relatively limited statistical sample, given the time window analyzed. I also think that much of the statistical information in the figures should be reproduced in a separate table for easier reference.**

**Second, another remark is the role MODIS ACAOD plays in the storytelling: although a separate section is devoted to it in the text, too little consideration is given to this dataset. This is from the abstract, in which MODIS is not even mentioned, throughout the conclusions. The MODIS dataset, which is a nice addition to the suite, should be discussed more because it carries valuable information.**

**Third (and most important remark so far).**

**The manuscript appears to be more of an account of a technical project than a scientific article. As it seems to me, this paper goes so far as it does in squeezing out information from the results presented so far. But a paper comes alive not by documenting a particular accomplishments, but by making connections with previous research. If this is not done, then also the interpretation of the results will be less informative.**

**While the differences between the models and measurements are clearly reported, I found no convincing attempt to go nor to the source of these**

**differences neither to scientifically pinpoint the outcome of these differences. All the more so when the authors themselves in the text indicate possible causes of discrepancy (i.e. PBL calculations, emission inventories and others) and also the possible consequences of these discrepancies (direct, semi-direct, indirect aerosol effects). As such the analysis feels shallow and much more could be inferred from the datasets at hands.**

**I believe is a reasonable task for such a comprehensive list of co-authors with their respective expertises.**

**Although I believe it is a choice of the authors with what content to populate a scientific article, I would appreciate more effort regarding two issues: (i) going as far as possible - and reasonably - to the source of AOD differences from models (e.g., looking at emission inventories and the treatment of air mass evolution with aerosols); (ii) identification - through measurements or other sources - of those instances where entrainment occurs, that is how much of the differences can be explained by (imperfect mixing). suboptimal representation of aerosols AND clouds together? Do the lidar profiles provide additional information on the aerosol and cloud thermodynamic phases (along the vertical) that might be of interest to the ESMs?**

**I want to be even clearer on this point: I do not expect the authors to dissect the source code of ESMs and analyze the respective parameterizations. Just as I do not expect them to quantify changes in radiative forcing depending on whether aerosols are detached, in contact, or inside clouds. This would be a topic for a separate study. What I would expect is an analysis - at least a qualitative one, based on both the abundant existing literatures and original reasoning - of the two aspects mentioned above.**

**And this should then be reported in a section called "Discussion", right before "Summary and conclusions" (which reads just as a repetition of the abstract).**

**I could request "minor revisions" and the paper might even be fine as it is, although scientifically thin. But I am requesting "major revisions" just to make sure that my comments are considered to augment the interpretation and to warrant the full scientific exploitation of the results obtained so far.**

We summarized the statistics of all the scatter plots in Table 2 for easier reference to readers. We also made connections between our results and previous studies by referencing related work such as Shinozuka et al. (2020) and Doherty et al. (2022) as shown in Section 3.4.

We need to emphasize that MODIS ACAOD is useful for qualitative assessment of the aerosol plume patterns rather than the magnitude of ACAOD. In Chang et al. (2021), we showed that this ACAOD product tends to be higher than the aircraft AOD

measurements and greatly depends on the assumed aerosol absorption (i.e., single scattering albedo). In the abstract and in the conclusion, we added the following:

"ALADIN and GEOS-Chem show similar in aerosol plume patterns when compared to the above-cloud aerosol product from the Moderate Resolution Imaging Spectroradiometer (MODIS) during September 2016, but none of the models show a similar above-cloud plume patterns as MODIS does in August 2017."

Regarding the first issue on sources of AOD differences, we significantly expanded Table 1 by elaborating on how emissions, transport and deposition processes are handled in each model. We also added a section called "Discussion on model deficiencies for future investigations" that addresses model deficiencies that may explain the errors on the estimated AODs for future investigations. This section provides a starting point for additional modeling studies to explore the impact of each factor on estimated AODs. Here is the additional section in the text:

"ESMs are complex and nonlinear systems, so AOD errors are likely caused by numerous factors. Identifying the exact causes of AOD biases is challenging and entails a detailed examination of model source codes. Here, we present aspects of the models that may explain their biases in simulated AOD relative to those measured by airborne lidar, which establishes a starting point for a future in-depth investigation. The assimilation of clear-sky MODIS AODs in the two assimilation systems (i.e., GEOS-FP and MERRA-2) may explain their better performance compared to other models in simulating AODs, especially in August 2017. Despite a lack of MODIS clear-sky AOD retrievals over regions with expansive cloud presence, such as in the austral spring of the SE Atlantic, AOD assimilation is still beneficial for minimizing AOD errors in ESMs. The mean and median AOD and the AOD fraction in the FT in WRF-FINN generally agree well with those from aircraft measurements. WRF-FINN is also the only model in this study that includes a plume rise parameterization. The importance of the inclusion of a plume rise model for simulating high AODs in this region is unclear since fire emissions in southern Africa already take place at elevated altitudes. Nonetheless, the smoke top heights in the remaining models generally agree with those from lidar measurements (Shinozuka et al., 2020).

The rate of primary organic aerosol (POA) removal and the secondary organic aerosol (SOA) production influences the simulated AOD (Hodzic et al., 2020). For example, the negligible production of SOA in WRF-CAM5, GEOS-FP, MERRA-2, and ALADIN may be contributing to a low bias in simulated AOD. For GEOS-Chem, GEOS-FP and MERRA-2, their aerosol optical properties are assumed to be fixed and do not account for particle evolution during transport. Even though the production of SOA is introduced in the other models, the assumed processes may be oversimplified such that its production is based on precursors at a fixed time-scale without a detailed consideration for chemistry. Moreover, these models do not treat photochemical loss of SOA as shown by its excessive OC according to Shinozuka et al. (2020). Errors in the treatment of aerosol

hygroscopicity may also play a crucial role in the aerosol evolution and subsequent AOD biases. Although the AOD fraction in the FT in WRF-CAM5 has a good agreement with lidar measurements, Shinozuka (2020) found that the PBL height of this model was a few hundred meters higher than that in lidar cloud-top measurements in September 2016, possibly leading to overactive entrainment and aerosol removal. While the selection of emission inventory alone impacts simulated AODs (Pan et al. 2020), the use of monthly emission inventory in both EAM-E3SM and ALADIN instead of diurnally-varied emissions as in other models could further be responsible for some of the errors. These deficiencies suggest that AOD errors in each model are likely driven by multiple factors, and a more in-depth model-specific analysis would be needed to investigate model deficiencies that leverages multiple degrees of freedom."

Regarding the second issue on identification through measurements, lidars cannot provide the location of entrainment but they can identify the contact location between aerosol and cloud with careful setting of extinction thresholds. As noted in the manuscript, we only examined AOD in layers without clouds due to enhanced AOD by humidification within clouds (though statistically indistinguishable at the 0.05 significance level). The HSRL-2 retrievals include aerosol type, which was mainly biomass burning smoke aerosol during the NASA ORACLES field campaign. The lidar can distinguish ice from liquid clouds through depolarization ratios. During the field campaign period, cloud regimes are mostly stratocumulus liquid clouds under the influence of the sub-tropical high pressure system. Some mid-level clouds (cloud top heights between 3 – 8 km) were observed from the lidar, which are supercooled liquid clouds. The cloud-top height from lidar allowed us to evaluate the modeled PBL height, which were thoroughly presented in the supplementary materials.

**Specific comments**

**P2 L71: the authors may want to be more precise with the reference about cloud lifetime and add the standard Albrecht's study (https://www.science.org/doi/10.1126/science.245.4923.1227)**

Done

**P8 L267-269: is it really true that varying the grid cell size affects the standard deviation only? How do you know this is a fact? I would expect that also the mean AOD will be affected. More precisely, finer cells have higher means than coarser cells. As a consequence, this will impact the comparison and the vertical partition of aerosol loading.**

We produced the scatter plots at various grid resolutions (0.25, 1, and 2.5 degrees) for 3 of the models as shown in Figure R1. The remaining models have high agreement across various grid resolutions as WRF-CAM5, which is not shown for brevity. As shown in Figure R1, grid resolution mainly affects standard deviation and very little on the other statistics except for the FE in ALADIN (dropped by 0.09 when changing from 0.25- to 2.5-degree).

[Figure]

Figure R1. Scatter plots comparing full-column AOD (at 550 nm) from EAM-E3SM (top row), ALADIN (middle row), and WRF-CAM5 (bottom row) to HSRL-2 clear-sky AOD during the September 2016 deployment of the ORACLES field experiment. The grid resolution changes from the left column to the right column for 0.25-degree, 1-degree, and 2.5-degrees

The last paragraph of Section 2.7 now include the following:

"In the second analysis, we evaluate the models' performances using various statistical metrics. We aggregate modeled and aircraft AODs to 1° grid resolution, which is approximately the median native grid resolution of the ESMs that we examine in this study. The Spearman's Rank correlation coefficient is used instead of the Pearson's linear correlation coefficient since the former is statistically less sensitive to outliers (Sayer et al., 2019; Sayer, 2020). We also evaluate the *RMSE*, the fractional error (*FE*), and the fractional bias (*FB*):

$$FE = \frac{2}{N}\sum \frac{|modeled\ AOD - observed\ AOD|}{(modeled\ AOD + observed\ AOD)} , (3)$$

$$FB = \frac{2}{N}\sum \frac{(modeled\ AOD - observed\ AOD)}{(modeled\ AOD + observed\ AOD)} , (4)$$

where $N$ is the sample size. Note that $FB$ is similar to the relative mean bias reported by Shinozuka et al. (2020b) except for the addition of the modeled values in the denominator and the factor of two outside the summation. Typically, up to 100 points of aircraft data are averaged into a 1° grid box. Varying the aggregated grid resolution mainly affects standard deviations and has a very minor influence on other statistics such as correlations and root mean square error ($RMSE$). The $FE$ and $FB$ for each model agrees to within 0.04 of their respective value when the data are aggregated between grid resolutions of 0.25° to 2.5°, except for ALADIN where its $FE$ decreases by 0.09 in going from 0.25° to 2.5° grid resolution."

The insensitivity to grid resolution results because the mean AOD could either increase or decrease when aggregating AOD at finer resolution. Although one grid may have higher AOD than a mean AOD over a coarser resolution, the AOD in neighboring grid cells may have lower value and vice versa. The long range transport of the regional haze also tends to be more uniform in aerosol loading, so AOD gradients in the models are generally smaller for the models with finer grid resolution.

**P12 L-378-380 (and P7 L225-226): How can MODIS ACAOD report higher AODs while the reported mean biases are negative?**

The values should be positive instead of negative. Thank you for identifying this error.

**P13 L 417-418: "A deeper analysis of biases in model processes than is possible through the AOD comparisons presented here is essential in order to understand the cause of model biases."**

**First, check the wording (than). Second, I understand that the results presented here are a first-order assessment of model performances against in-situ and spaceborne observations. As such, the authors suggest the examination of those assumptions and paramterizations leading to the found biases. While the statistics and the presentation of the found biases are sufficiently clear and exhaustive, I miss then the takeaway. Isn't an author's task to reasonably pinpoint the error sources instead of leaving the question open?**

We removed that statement because we addressed model deficiencies that may explain the factors for AOD errors for future investigations. As mentioned in the general comments, the discussions will provide a starting point for the needed step of additional model studies to explore how each factor is affecting AOD.

**P14 L 444-458: The short discussion about the nature of aerosol radiative forcing, while correct, feels premature or feels like a natural conclusion of the study for others to be answered.**

We agree that these statements may be premature. Our intention to briefly discussing aerosol radiative forcing is to inform readers on the broader implications of this work and tie back to the introduction. We purposefully provide general statements on different aerosol radiative forcing without discussing the signs and magnitude under various scenarios since it would require a lengthy discussion to comprehensively depict a topic that is well beyond the scope of this work.

**Except for the wording that could be stylistically improved ("In conditions where ... play roles". Seemingly redundant), I am left with the question of where and how the authors ever touched in their main text upon the presence of layered clouds, the entrainment of aerosols, and ensuing change of thermodynamic phase and change in extinction profile.**

We replaced "play roles" with "are significant". The lidar can identify multilayered cloud provided that the top cloud layer does not completely attenuate the backscatter signal. We added this statement in the introduction "mid-level clouds can also be present and be in contact with aerosols above low-level liquid clouds (Adebiyi et al., 2020)". As indicated earlier, lidars cannot provide the location of entrainment but they can identify the contact location between aerosol and cloud with careful setting of extinction thresholds. Models have multi-layer clouds, but we only considered AOD of layers that are cloud-free. Regarding entrainment, we mentioned in the Discussion section "Although the AOD fraction in the FT in WRF-CAM5 has a good agreement with lidar measurements, Shinozuka (2020) found that the PBL height of this model was a few hundred meters higher than that in lidar cloud-top measurements in September 2016, possibly leading to overactive entrainment and aerosol removal". Doherty et al. (2022) conducted a comprehensive analysis of differences in extinction profiles for the same WRF-CAM5 and GEOS-FP models used in this study. In section 3.4, we added "Doherty et al. (2022) noted that extinction profiles of WRF-CAM5 and GEOS-FP generally tend to be lower than those measured by the HSRL-2. They also found that extinction profiles are more vertically diffuse with weaker vertical gradients than the lidar measurements.

---

## Author Comment (AC2)

**Response to reviewers for Chang et al. (acp-2022-496)**

We thank the reviewers for their constructive comments and suggestions. We are glad to hear that the reviewers find this work well-written and suitable for consideration to be published in ACP. Our responses are in bold-free texts and the reviewers' comments are in bold texts.

Reviewer 2

**This well written manuscript does a thorough evaluation of the ability of five ESMs to represent the partitioning of the Aerosol Optical Depth (AOD) between the free troposphere and the marine boundary layer over the southeast Atlantic. It takes advantage of the instruments that were deployed to measure aerosol properties during the two campaigns ORACLES 2016 and ORACLES 2017.**

**The description of both the models under study and of the instruments used to retrieve the AODs at different heights is well conducted but it leaves the reader wandering about what we have learned about the processes that explaining the discrepancies between model and observations that are described in the conclusions. A complete study requires to consider what are the sources of uncertainties in these models and how they differ from one another both in how they parameterize their processes and also on how they consider the different source inventories for fires that emit these aerosols. A much stronger paper would emerge if the authors took up the (difficult) task to explain why models either underestimate either overestimate the AODs and what processes the modelers should focus on to improve on the results. This extra work will make up for a much better paper.**

We thank the reviewer for the constructive comments and suggestions. We are glad to hear that the reviewer finds this work well-written. We significantly expanded Table 1 by elaborating on how emissions, transport and deposition processes are handled in each model. We also added a section called "Discussion on model deficiencies for future investigations" that addresses model deficiencies that may explain the factors for AOD errors for future investigations. The discussions will provide a starting point for the needed step of additional model studies to explore how each factor is affecting AOD. Here is the additional text for that section:

"ESMs are complex and nonlinear systems, so AOD errors are likely caused by numerous factors. Identifying the exact causes of AOD biases is challenging and entails a detailed examination of model source codes. Here, we present aspects of the models that may explain their biases in simulated AOD relative to those measured by airborne lidar, which establishes a starting point for a future in-depth investigation. The assimilation of clear-sky MODIS AODs in the two assimilation systems (i.e., GEOS-FP and MERRA-2) may explain their better performance compared to other models in simulating AODs, especially in August 2017. Despite a lack of MODIS clear-sky AOD retrievals over regions with expansive cloud presence, such as in the austral spring of the SE Atlantic, AOD assimilation is still beneficial for minimizing AOD errors in ESMs.

The mean and median AOD and the AOD fraction in the FT in WRF-FINN generally agree well with those from aircraft measurements. WRF-FINN is also the only model in this study that includes a plume rise parameterization. The importance of the inclusion of a plume rise model for simulating high AODs in this region is unclear since fire emissions in southern Africa already take place at elevated altitudes. Nonetheless, the smoke top heights in the remaining models generally agree with those from lidar measurements (Shinozuka et al., 2020).

The rate of primary organic aerosol (POA) removal and the secondary organic aerosol (SOA) production influences the simulated AOD (Hodzic et al., 2020). For example, the negligible production of SOA in WRF-CAM5, GEOS-FP, MERRA-2, and ALADIN may be contributing to a low bias in simulated AOD. For GEOS-Chem, GEOS-FP and MERRA-2, their aerosol optical properties are assumed to be fixed and do not account for particle evolution during transport. Even though the production of SOA is introduced in the other models, the assumed processes may be oversimplified such that its production is based on precursors at a fixed time-scale without a detailed consideration for chemistry. Moreover, these models do not treat photochemical loss of SOA as shown by its excessive OC according to Shinozuka et al. (2020). Errors in the treatment of aerosol hygroscopicity may also play a crucial role in the aerosol evolution and subsequent AOD biases. Although the AOD fraction in the FT in WRF-CAM5 has a good agreement with lidar measurements, Shinozuka (2020) found that the PBL height of this model was a few hundred meters higher than that in lidar cloud-top measurements in September 2016, possibly leading to overactive entrainment and aerosol removal. While the selection of emission inventory alone impacts simulated AODs (Pan et al. 2020), the use of monthly emission inventory in both EAM-E3SM and ALADIN instead of diurnally-varied emissions as in other models could further be responsible for some of the errors. These deficiencies suggest that AOD errors in each model are likely driven by multiple factors, and a more in-depth model-specific analysis would be needed to investigate model deficiencies that leverages multiple degrees of freedom."